# GENERALIZED GROUP DATA ATTRIBUTION

## ABSTRACT

Data Attribution (DA) methods quantify the influence of individual training data points on model outputs and have broad applications such as explainability, data selection, and noisy label identification. However, existing DA methods are often computationally intensive, limiting their applicability to large-scale machine learning models. To address this challenge, we introduce the Generalized Group Data Attribution (GGDA) framework, which computationally simplifies DA by attributing to groups of training points instead of individual ones. GGDA is a general framework that subsumes existing attribution methods and can be applied to new DA techniques as they emerge. It allows users to optimize the trade-off between efficiency and fidelity based on their needs. Our empirical results demonstrate that GGDA applied to popular DA methods such as Influence Functions, TracIn, and TRAK results in upto **10x-50x** speedups over standard DA methods while gracefully trading off attribution fidelity. For downstream applications such as dataset pruning and noisy label identification, we demonstrate that GGDA significantly improves computational efficiency and maintains effectiveness, enabling practical applications in large-scale machine learning scenarios that were previously infeasible.

## 1 INTRODUCTION

In the era of data-driven machine learning, the impact of training data on model performance has become increasingly evident. As such, the ability to discern and quantify the influence of individual training examples on model behavior has emerged as a critical area of research. Data Attribution (DA) methods have risen to powerful tools designed to address this need, which identify the influence of each training data point for specific test predictions, offering insights into the complex relationships between training data and model behavior (Koh & Liang, 2017; Ghorbani & Zou, 2019; Pruthi et al., 2020; Schioppa et al., 2022; Jia et al., 2021; Kwon & Zou, 2022; Ilyas et al., 2022; Park et al., 2023; Wang & Jia, 2023). The applications of DA are far-reaching and diverse, encompassing crucial tasks such as model debugging, strategic data selection, and the identification of mislabeled instances. These capabilities hold immense potential for enhancing model reliability.

However, existing DA techniques face significant challenges that impede their practical application. Foremost among these is the issue of computational efficiency. These methods incur prohibitive computational costs, for example, by requiring training a large number of models in the order of the training data size, which is computationally intractable for large-scale machine learning models (Ghorbani & Zou, 2019; Ilyas et al., 2022). On the other hand, many existing methods struggle with attribution fidelity, often failing to faithfully represent the true impact of training data on model outcomes (Basu et al., 2020a). This lack of fidelity can lead to misinterpretations and potentially misguided model development and deployment decisions. These efficiency and fidelity limitations combined have severely restricted the widespread adoption and utility of DA techniques in large-scale applications.

To address these critical issues, we propose a novel approach called Generalized Group Data Attribution (GGDA). This method shifts the paradigm from attributing influence to individual data points to considering groups of data points collectively. By aggregating data into meaningful groups, GGDA substantially reduces the computational overhead of attribution, making it particularly well-suited for large-scale applications where traditional DA methods falter. Importantly, this improvement in efficiency comes at little cost to attribution fidelity. Overall, this framework improves the scalability of individual-point attribution techniques while maintaining and, in some cases, improving the at-

tribution fidelity. GGDA also offers a unique advantage in allowing users to fine-tune the trade-off between efficiency and fidelity through adjustable group sizes. This adaptability empowers practitioners to optimize the balance between computational efficiency and attribution fidelity based on their specific requirements. While other approaches have been proposed to accelerate DA methods, such as subsampling and gradient approximations (Guo et al., 2021; Schioppa et al., 2022), these are often method-specific and focus on optimizing computational aspects of existing DA algorithms. In contrast, GGDA proposes a fundamentally novel attribution paradigm, allowing the acceleration of arbitrary DA methods.

To validate the effectiveness of GGDA, we conduct extensive experimental evaluations across various datasets (HELOC, MNIST, CIFAR10, QNLI), model architectures (ANN, ResNet, BERT), and DA methods (Influence Functions, TRAK, TracIn). Our results show that: (1) GGDA significantly improves the computational efficiency of DA methods, resulting in upto **10x-50x** speedups depending on the setting, (2) GGDA favourably trades off fidelity with efficiency compared to standard DA approaches, and (3) For downstream applications such as dataset pruning and noisy label identification, GGDA outperforms standard DA methods, while maintaining significant speedups.

## 2 RELATED WORK

**Data Attribution** quantify the influence of training samples on model predictions. One type of DA is retraining-based, which systematically retrain models with varying subsets of training data to measure sample influence. Leave-One-Out (LOO) influence Tukey (1958) measures the prediction difference between models trained on the full dataset and those trained with one sample removed. Data Shapley (Ghorbani & Zou, 2019; Jia et al., 2019) extends this concept by considering all possible subsets of training data, providing a more comprehensive measure of sample importance. DataModels (Ilyas et al., 2022) attempts to learn model predictions for each subset of the training dataset, requiring significant computational resources. While these methods can capture complex data interactions, they are computationally expensive due to the need for multiple model retrainings. Another type of gradient-based method offers more scalable solutions by providing closed-form attribution scores using gradients. Influence functions (Koh & Liang, 2017) approximate the effect of upweighting a training sample on the loss function, providing a foundation for many subsequent studies. TracIn (Pruthi et al., 2020) traces loss changes on test points during the training process, offering insights into sample influence throughout model training. TRAK (Park et al., 2023) employs the neural tangent kernel with random projection to assess influence, presenting a novel approach to attribution. These gradient-based methods significantly reduce computational costs compared to retraining-based approaches but may suffer from performance degradation on non-convex neural networks due to their reliance on convexity assumptions and Taylor approximations.

**Grouping in Data Attribution.** Several prior works considered attributing to groups, emphasizing different aspects of the problem than ours. For example, Koh et al. (2019) examines influence functions when applied to large groups of training points, and their empirical analysis revealed a surprisingly strong correlation between the predicted and actual effects of groups across various datasets. While it focuses primarily on linear classification models, we particularly focus on large-scale non-linear models and study the fidelity and computational efficiency of group attribution. Basu et al. (2020b) proposed a second-order influence function to identify influential groups of training samples better. While their estimator results in improved fidelity estimates, their runtime is also strictly worse than first-order influence methods. On the other hand, our focus is on improving computational efficiency, even at slight costs to fidelity. To overcome the computational cost of Data Shapley, Ghorbani & Zou (2019) also briefly mentioned the idea of grouping data points to manage large datasets more effectively as an experimental design choice. On the other hand, our work studies group attribution more systematically and shows that (1) it is possible to generalize both retraining-based and gradient-based attribution methods to the group setting and (2) it demonstrates experimentally the superiority of group approaches.

**Data Attribution Acceleration.** Various approaches have been proposed to accelerate data attribution methods, including subsampling (Guo et al., 2021) and better gradient/Hessian approximations (Schioppa et al., 2022). These techniques focus on optimizing existing attribution algorithms to improve computational efficiency. While valuable, these methods are orthogonal to our GGDA framework, which fundamentally alters the attribution paradigm by considering groups of data points. As

such, GGDA can potentially be combined with these acceleration techniques to achieve even greater efficiency gains, highlighting the complementary nature of our approach.

## 3 GENERALIZED GROUP DATA ATTRIBUTION FRAMEWORK

This section introduces the formal definitions of Data Attribution (DA) and our proposed Generalized Group Data Attribution (GGDA) frameworks. We examine the limitations of the conventional DA methods, which attributes importance to individual training points, and show that they often result in prohibitive computational costs. We then present the rationale behind the development of GGDA as a solution to these challenges. We begin by establishing the notations and definitions.

**Notation.** Given a dataset of $n$ training data points $\mathcal{D} = \{(\mathbf{x}_0, \mathbf{y}_0), (\mathbf{x}_1, \mathbf{y}_1), ..., (\mathbf{x}_n, \mathbf{y}_n)\}$ in the supervised learning setting[1], and a learning algorithm $\mathcal{A}(\mathcal{D}) : 2^{\mathcal{D}} \to \mathbb{R}^p$ that outputs weights $\theta \in \mathbb{R}^p$, where $f(\mathbf{x}_{\text{test}}; \theta) \in \mathbb{R}$ is the model prediction function for a given test input $\mathbf{x}_{\text{test}}$, we define a counterfactual dataset $\mathcal{D}_S = \mathcal{D} \setminus \{(\mathbf{x}_i, \mathbf{y}_i; i \in S)\}$ as the training dataset with points in $S$ removed. We use $\theta_S \sim \mathcal{A}(\mathcal{D}_S)$ to denote the model weights trained on $\mathcal{D}_S$.

**Definition 1** *(Data Attribution) Given a learning algorithm $\mathcal{A}$, dataset $\mathcal{D}$ and test point $\mathbf{x}_{test}$, data attribution for the $i^{th}$ training point $(\mathbf{x}_i, \mathbf{y}_i)$ is a scalar estimate $\tau_i(\mathcal{A}, \mathcal{D}, \mathbf{x}_{test}) \in \mathbb{R}$ representing the influence of $(\mathbf{x}_i, \mathbf{y}_i)$ on $\mathbf{x}_{test}$.*

To ground our discussion with a concrete example, we first introduce the conceptually most straightforward DA method, namely the Leave-One-Out (LOO) estimator. This computes the influence of individual training points, by the average prediction difference between the model trained on the original dataset, and the one trained on the dataset with a single training point excluded.

**Definition 2** *(Leave-One-Out estimator) The LOO estimator of a learning algorithm $\mathcal{A}$, dataset $\mathcal{D}$ and test point $\mathbf{x}_{test}$, w.r.t. to training point $(\mathbf{x}_i, \mathbf{y}_i)$ is given by $\tau_i(\mathcal{A}, \mathcal{D}, \mathbf{x}_{test}) = \mathbb{E}_{\theta \sim \mathcal{A}(\mathcal{D})} f(\mathbf{x}_{test}; \theta) - \mathbb{E}_{\theta_i \sim \mathcal{A}(\mathcal{D} \setminus \{(\mathbf{x}_i, \mathbf{y}_i)\})} f(\mathbf{x}_{test}; \theta_i)$*

The LOO estimator forms the conceptual basis for methods in the DA literature. For example, the influence function (Koh & Liang, 2017) based methods are gradient-based approximations of the LOO estimator. On the other hand, methods like datamodels (Ilyas et al., 2022) and TRAK (Park et al., 2023) aim to generalize beyond LOO estimators by estimating importance of training points by capturing the average effect of excluding larger subsets that include that training point. However, the underlying paradigm of measuring the importance of individual training points $(\mathbf{x}_i, \mathbf{y}_i)$, for a given test point $\mathbf{x}_{\text{test}}$, remains common among DA methods. However, as we shall now discuss, this paradigm of data attribution has several critical drawbacks.

**Drawback: Attribution Computation Scales with Data Size.** A critical challenge in data attribution is its substantial computational cost. From an abstract perspective, the definition of DA necessitates computing attributions for each training point individually, resulting in computational requirements that scale with the size of the training dataset $n$. This scaling becomes prohibitive in modern learning settings with large $n$. The computational burden can be even more severe in practice. For instance, the LOO estimator demands training at least $n + 1$ models, with this number increasing significantly in non-convex settings due to Monte Carlo estimation for the expectation in Definition 2 (Jia et al., 2019). Gradient-based methods, such as influence functions, aim to reduce attribution computation by avoiding retraining. However, these methods still require (1) computation of per-training-sample gradients w.r.t. parameters and (2) computation of the Hessian inverse of models parameters. Both these steps are computationally intensive, as (1) requires $n$ forward and backward passes, which is prohibitive when $n$ is in the billions, and (2) requires manipulating the Hessian matrix whose dimensions are $p \times p$ for a model $\theta \in \mathbb{R}^p$, which is again intractable when $p$ is in the billions.

**Drawback: Pointwise Attribution may be Ill-Defined.** Another issue with DA is that estimates of individual training point importance may not always be meaningful. For example, recall that the LOO estimator involves repeatedly training models on *nearly identical* training sets, with merely a single point missing. However, for training data involving billions of data points, a single missing

---

[1]This can be easily extended to unsupervised learning

example out of billions is unlikely to lead to any statistically significant change in model parameters or outputs. This intuition is captured in the machine learning literature via properties such as (1) stability of learning algorithms (Elisseeff et al., 2005), or (2) differential privacy (Dwork, 2008). These capture the intuition that the models output by learning algorithms must not change when individual points are removed, and these have been linked to (1) model generalization for the case of stability and (2) privacy guarantees on the resulting model. Thus, when we employ learning algorithms that satisfy these conditions, the LOO estimator, which is the conceptual basis for many DA methods, may be ill-defined.

**Drawback: Focus on Individual Predictions Limits Scope.** A primary motivation for DA is to *explain* individual model predictions via their training data. However, there exist many applications beyond explainability of individual predictions that require reasoning about the relationship between model behaviour and training data. For instance, we may wish to identify training points that cause models to be inaccurate, non-robust, or unfair, all of which are "bulk" model properties as opposed to individual predictions. One way to adapt DA to analyze such bulk model properties is to analyze pointwise the test points that are inaccurate or non-robust, for example, but this is undesirable for two reasons. First, many model properties, such as fairness and f-measure, are non-decomposable (Narasimhan et al., 2015), i.e., they cannot be written in terms of contributions of individual test points. Second, even with decomposable metrics, it may be computationally wasteful to identify important training instances for the bulk metric by aggregating results across test points instead of directly identifying important training data points for the metric itself. We are thus interested in developing tools to directly analyze the impact of training data on bulk model behavior.

To resolve these drawbacks, we propose to modify the DA framework by partitioning the training dataset into groups of training points and attributing model behaviour to these groups instead of individual data points. Similar data points affect models in similar ways, so it is meaningful to consider attribution to groups consisting of similar data points. When no such meaningful groups exist, we can define groups as singleton training points, subsuming the standard DA setting. Note that our expanded scope of DA can handle not just individual predictions but arbitrary functions of the model parameters, which we call "property functions". This can help efficiently reason about bulk model properties such as accuracy, robustness, and fairness. When we wish to attribute individual model predictions, we can simply set the property function to be equal to the loss for an individual test sample, thus subsuming the case of standard DA. We now formally define these two new components: groups and property functions, before using these to define our GGDA framework.

**Group** $\mathcal{Z}$: Given a dataset $\mathcal{D}$, we define a partition of the dataset into $k$ groups such that $\mathcal{Z} = \{\mathbf{z}_0, \mathbf{z}_1, ...\mathbf{z}_k\}$, where each $\mathbf{z}_j$ is a set of inputs $\mathbf{x}_i$ such that $\mathbf{z}_j \cap \mathbf{z}_i = \emptyset$ for $i \neq j$, and $\bigcup_i \mathbf{z}_i = \mathcal{D}$. In general, the cardinality of each group $|\mathbf{z}_i|$ can be different. When the group sizes = 1, this reverts to the usual case of attributing to individual training data points.

**Property function** $g$: Given a model with weights $\theta \in \mathbb{R}^p$, we define a property function $g(\cdot) : \mathbb{R}^p \rightarrow \mathbb{R}$. Property functions generalize (1) pointwise functions like log probability of a given test point, (2) aggregate functions like accuracy on a test set. Given these two quantities, we are ready to define GGDA.

**Definition 3** *(Generalized Group Data Attribution) Given a learning algorithm $\mathcal{A}$, dataset $\mathcal{D}$ with groups $\mathcal{Z}$ and property function $g$, GGDA wrt to the $j^{th}$ group $\mathbf{z}_j$ estimates $\tau_j(\mathcal{A}, \mathcal{D}, \mathcal{Z}, g) \in \mathbb{R}$ representing the influence of group $\mathbf{z}_j$ on model property $g$*

To ground this with a concrete example, we present the GGDA variant of the LOO estimator below. The idea is very similar to the usual LOO estimator, except here that we leave the $j^{th}$ group out, and measure the resulting change in the property function $g$.

**Definition 4** *The (GGDA LOO) estimator of a learning algorithm $\mathcal{A}$, dataset $\mathcal{D}$, groups $\mathcal{Z}$ and property function $g$, wrt to the $j^{th}$ group $\mathbf{z}_j$ is given by $\tau_j(\mathcal{A}, \mathcal{D}, \mathcal{Z}, g) = \mathbb{E}_{\theta \sim \mathcal{A}(\mathcal{D})} g(\theta) - \mathbb{E}_{\theta_j \sim \mathcal{A}(\mathcal{D} \setminus Z_j)} g(\theta_j)$*

Comparing this to the usual LOO attributor, we observe that the GGDA variant requires training of $k+1$ models as opposed to $n+1$ models, which is advantageous when $k \ll n$. In summary, the main advantage of GGDA methods is that they scale as the number of groups $k$, whereas DA methods scale with the number of data points $n$. While extending LOO was fairly simple, it is, in general,

non-trivial to extend arbitrary DA methods to GGDA methods, particularly in a way that makes the computation scale with the number of groups $k$, reflecting the advantage that group attribution offers. For example, while Koh et al. (2019) study group influence functions, they compute these group influences by summing the influences of individual training points, thus retaining the computational cost of standard DA. In the next section, we shall propose GGDA variants of popular DA methods in a manner that exactly improves their computational efficiency.

# 4 GENERALIZING EXISTING DATA ATTRIBUTION METHODS TO GROUPS

In this section, we show how to generalize existing DA algorithms to the GGDA setting, retaining the computational benefits of attribution in groups. In particular, we focus on deriving GGDA variants of gradient-based influence methods, given their inherent computational superiority over re-training-based methods. To facilitate the derivation of the group setting, we unify two gradient-based DA methods — TRAK and TracIn — as instances of influence function methods, even as this differs with the original motivation presented in these papers. To derive these group variants, there are two principles we follow:

- the GGDA variants must scale with $k$, the number of groups, and not $n$, the number of datapoints;
- the GGDA variants must subsume the ordinary DA variants when groups are defined as singleton datapoints, and the property function as the loss on a single test point.

With these principles in mind, we start our analysis with influence functions.

**Generalized Group Influence Functions.** Given a loss function $\ell \in \mathbb{R}^+$, the Influence Function data attribution method (Koh & Liang, 2017) is given by:

$$\tau_{\inf}(\mathcal{A}, \mathcal{D}, \mathbf{x}_{\text{test}}) = \underbrace{\nabla_\theta \ell(\mathbf{x}_{\text{test}}; \theta)^\top}_{\mathbb{R}^{1 \times p}} \underbrace{H_\theta^{-1}}_{\mathbb{R}^{p \times p}} \underbrace{\left[ \nabla_\theta \ell(\mathbf{x}_0; \theta); \nabla_\theta \ell(\mathbf{x}_1; \theta); ... \nabla_\theta \ell(\mathbf{x}_n; \theta) \right]}_{\mathbb{R}^{p \times n}} \quad \theta \sim \mathcal{A}(\mathcal{D})$$

The primary computational bottlenecks for computing influence functions involve (1) computing the Hessian inverse, and (2) computing the $n$ gradients terms, which involve independent $n$ forward and backward passes. While several influence approximations (Guo et al., 2021; Schioppa et al., 2022) focus on alleviating the bottleneck due to computing the Hessian inverse, here we focus on the latter bottleneck of requiring order $n$ compute. While efficient computation of per-sample gradients has received attention from the machine learning community (Bradbury et al., 2018) via tools such as vmap, the fundamental dependence on $n$ remains, leading to large runtimes. In the Appendix A.1, we show that due to the linearity of influence functions, the influence of groups is given by the sum of influences of individual points within the group, which is consistent with the analysis in Koh et al. (2019). Overall, we show that the corresponding GGDA generalization is:

$$\tau_{\inf}(\mathcal{A}, \mathcal{D}, \mathcal{Z}, g) = \underbrace{\nabla_\theta g(\theta)^\top}_{\mathbb{R}^{1 \times p}} \underbrace{H_\theta^{-1}}_{\mathbb{R}^{p \times p}} \underbrace{\left[ \nabla_\theta \ell(\mathbf{z}_0; \theta); \nabla_\theta \ell(\mathbf{z}_1; \theta); ... \nabla_\theta \ell(\mathbf{z}_k; \theta) \right]}_{\mathbb{R}^{p \times k}} \quad \theta \sim \mathcal{A}(\mathcal{D}) \quad (1)$$

$$\text{where} \quad \ell(\mathbf{z}_i) := \sum_{\mathbf{x} \in \mathbf{z}_i} \ell(\mathbf{x})$$

From equation 1, we observe that the corresponding group influence terms involve $k$ batched gradient computations instead of $n$ per-sample gradient computations, where all $\sim n/k$ points belonging to each group are part of the same batch. Experimentally, we observe that computing a single batched gradient is roughly equivalent in runtime to computing individual per-sample gradients. This combined with the fact that typically $k \ll n$, leads to considerable overall runtime benefits for GGDA influence functions.

However, we also note that there exists a lower limit on the number of groups $k$ where this analysis holds. Practically, hardware constraints result in maximum allowable value for batch size, which we denote by $b$. If $b > n/k$, then points in each group can be fit into a single batch, leading to the computational advantage mentioned above. Thus a practical guide to set the number of groups is using $k \gtrsim n/b$ to obtain the maximum computational benefit. Beyond this value, using GGDA

influence provides a constant factor speedup of $b$, that still nevertheless results in significant real-world speedups. We use this key insight regarding speeding up group influence functions for two more methods: TracIn and TRAK, by viewing them as special cases of influence methods.

**TracIn as a Simplified Influence Approximation.** Pruthi et al. (2020) propose an influence definition based on aggregating the influence of model checkpoints in gradient descent. Formally, TracIn can be described as follows:

$$\tau_{\text{identity-inf}}(\theta, \mathcal{D}, \mathbf{x}_{\text{test}}) = \nabla_\theta \ell(\mathbf{x}_{\text{test}}; \theta)^\top \left[ \nabla_\theta \ell(\mathbf{x}_0; \theta); \nabla_\theta \ell(\mathbf{x}_1; \theta); ... \nabla_\theta \ell(\mathbf{x}_n; \theta) \right] \tag{2}$$

$$\tau_{\text{tracein}}(\mathcal{A}, \mathcal{D}, \mathbf{x}_{\text{test}}) = \sum_{i=1}^{T} \tau_{\text{identity-inf}}(\theta_i, \mathcal{D}, \mathbf{x}_{\text{test}}) \quad \text{where } \{\theta_i | i \in [1, T]\} \text{ are checkpoints} \tag{3}$$

The core DA method used by TracIn is the standard influence functions, with the Hessian being set to identity, which drastically reduces computational cost. The GGDA of TracIn thus amounts to replacing the `identity-inf` component (Eq 4) with Eq 1, with the Hessian set to identity.

**TRAK as Ensembled Fisher Influence Functions** One of the popular techniques for Hessian approximation in machine learning is via the Fisher Information Matrix, and the empirical Fisher matrix. Some background material on these is presented in the Appendix A.3. The DA literature has also adopted these approximations, with Barshan et al. (2020) using influence functions with the Fisher information matrix, TRAK (Park et al., 2023) using the empirical Fisher estimate. While the TRAK paper itself views their estimate as a (related) Gauss-Newton approximation of the Hessian, we show in the Appendix A.4 that under the setting of the paper, this is also equivalent to using an empirical Fisher approximation. Atop the empirical Fisher approximation, TRAK also further reduces the computational cost by using random projections of gradients to compute the empirical Fisher. Given this, we denote TRAK as follows:

$$\tau_{\text{fisher-inf}}(\theta, \mathcal{D}, \mathbf{x}_{\text{test}}) = \nabla_\theta \ell(\mathbf{x}_{\text{test}}; \theta)^\top \hat{F}_\theta^{-1} \left[ \nabla_\theta \ell(\mathbf{x}_0; \theta); \nabla_\theta \ell(\mathbf{x}_1; \theta); ... \nabla_\theta \ell(\mathbf{x}_n; \theta) \right]$$

$$\tau_{\text{TRAK}}(\mathcal{A}, \mathcal{D}, \mathbf{x}_{\text{test}}) = \sum_{i=1}^{M} \tau_{\text{fisher-inf}}(\theta_i, \mathcal{D}, \mathbf{x}_{\text{test}}) \quad \text{where } \theta_i \sim \mathcal{A}(\mathcal{D}'), \mathcal{D}' \subset \mathcal{D}$$

Here, $\hat{F}_\theta$ is the empirical Fisher matrix, given by the summed outer product of per-sample gradients. Given that TRAK can be viewed as an aggregated influence function method, our technique for generalizing to group variants of influence functions directly applies here. Compared to vanilla influence functions, the only computational bottleneck that applies to empirical Fisher influence functions is the computation of per-sample gradients, which is also used to compute the empirical Fisher. Replacing usage of per-sample gradients with batched gradients, leads also to the following "batched" empirical Fisher approximation for a loss function $\ell$:

$$\hat{F}_\theta^{\text{batched}} = \sum_{\mathbf{z}} \nabla_\theta \ell(\mathbf{z}, \theta) \nabla_\theta \ell(\mathbf{z}, \theta)^\top \quad \text{where } \ell(\mathbf{z}, \theta) := \sum_{\mathbf{x} \in \mathbf{z}} \ell(\mathbf{x}, \theta)$$

When the groups are organized such that the gradients $\nabla_\theta \ell(\mathbf{x}_i)$ are aligned for all $i$ within a single group, then we can expect a small approximation error between the empirical Fisher and the batched empirical Fisher. Practically, this can occur if we pre-select groups based on their gradient similarity, i.e., performing k-means clustering on (approximations of) the gradient representations $\nabla_\theta \ell(\mathbf{x})$ of points. In summary, we have proposed generalized group extensions of three popular DA methods, such that (1) their computation scales are based on the number of groups rather than a number of data points, and (2) they subsume their DA variants. So far, we have not reasoned about the fidelity of these methods compared to DA, and we shall do so experimentally.

## 5 EXPERIMENTS

This section presents highlights from a comprehensive evaluation of our proposed GGDA framework. We conduct experiments on multiple datasets using various models to demonstrate the effectiveness and efficiency of our approach compared to existing data attribution methods. Full configuration details for datasets, models, attribution methods, grouping methods, and evaluations are listed in Appendix B, while the remainder of results across our grid of experiments is in Appendix C.

### 5.1 DATASETS AND MODELS

We evaluate across four diverse datasets encompassing tabular (HELOC), image (MNIST and CIFAR-10), and text data (QNLI). These datasets represent a range of task complexities, data types, and domains, allowing us to assess the generalizability of our approach across different scenarios. The models we fit on these datasets range from simple logistic regression (LR) to medium-sized artificial neural networks (ANNs, e.g., MLPs and ResNet) and pre-trained language models (BERT).

### 5.2 GROUPING METHODS

We employ several grouping methods to evaluate the effectiveness of our GGDA framework. For each grouping method, we also experiment with various group sizes to analyze the trade-off between computational efficiency and attribution effectiveness. The choice of grouping method and group size can significantly impact the performance of GGDA, as we subsequently demonstrate.

**Random** We uniformly randomly assign data points to groups of a specified size. This serves as our baseline method from which we may assess the impact of more sophisticated methods.

**K-Means** We use the standard K-means algorithm to cluster data points based on their raw features.

**Representation K-Means (Repr-K-Means)** We first obtain hidden representations of the data points from a model trained on the full training set. We then apply K-means clustering on these representations. This approach groups data points that are similar in the model's learned feature space, potentially capturing higher-level semantic similarities.

**Gradient K-Means (Grad-K-Means)** We compute the gradient of the loss with respect to the activations (NOT parameters) of the pre-classification layer of the model for each data point, and then apply K-means on the gradients. This approach groups instances with similar effects on the model's learning process.

### 5.3 EVALUATION METRICS

To illustrate the effectiveness of GGDA at identifying important datapoints, especially in comparison to DA, it is critical to have metrics that allow us to compare both on equal terms. To this end, we utilize the following metrics:

**Retraining Score (RS)** This metric quantifies the impact of removing a proportion of training data points on model performance. The process involves first identifying the most influential data points or groups using GGDA. A given percentile of the high-importance data points are then removed from the training set, and the model is retrained on the remaining data. We then compare the performance of this retrained model to the original model's performance on the test set. A significant decrease in performance (i.e., an increase in loss or decrease in accuracy) after removing influential points indicates that they were indeed important for the model's learning process. This metric provides a tangible measure of the effectiveness of GGDA in identifying highly useful training instances.

**Noisy Label Detection (AUC)** This metric quantifies the effectiveness with which GGDA methods can identify mislabelled training data points. This involves randomly flipping a proportion of the training set labels, and training a model on the corrupted dataset. We then examine the labels of the training data points that belong to groups with the lowest scores, and plot the change of the fraction of detected mislabeled data with the fraction of the checked training data, following Ghorbani & Zou (2019); Jia et al. (2021). The final metric is computed as the area under the curve (AUC).

## 5.4 RESULTS

In the following subsections, we present detailed evaluation results from across our range of datasets and models, discussing the performance of our GGDA framework in terms of retraining score, computational efficiency, and effectiveness in downstream tasks such as dataset pruning and noisy label detection. Our pipeline first trains a model on the full training set, before determining groups as described in Section 5.2. For each grouping method, we ablate extensively over a range of group sizes, beginning at 1 (standard DA) and increasing to 1024, in ascending powers of 2.

### 5.4.1 RETRAINING SCORE

To evaluate the retraining score metric, we compute *Test Accuracy* upon removal of 1%, 5%, 10%, and 20% of the most important points in the train set. For groups, this translates to sequentially removing the most important groups up until the desired number of data points (points are randomly selected from the final group in the case of overlap). Lower test accuracies are thus more favorable, indicating that the removed points were indeed important.

**Which Grouping Method to Use?** Before presenting our results, we first investigate the effect that the particular group arrangement has on GGDA methods, to select an appropriate grouping method. We plot a subset of group sizes that decreases from left to right as runtimes simultaneously increase, up unto the point where our GGDA framework subsumes traditional individual DA at group size 1. Among the grouping methods, we find that across the board, Grad-K-Means consistently demonstrates superior performance. Figure 1 illustrates this for the MNIST dataset (similarly conclusive plots for the remaining datasets and models are provided in Appendix C.1).

While random grouping necessarily provides orders of magnitude speedups in runtime, it is inferior at grouping together points that similarly impact model performance. For instance, the most important Random groups of size 64 consisted of a rough 50:50 split between points that had positive and negative individual attributions, while the most important Grad-K-Means group of size 64 consisted only of points with positive individual attributions (across all three attribution methods shown). In terms of preserving group integrity, the Grad-K-Means method outperforms other approaches by consistently grouping together data points that influence model performance in similar ways.

**Results across Removal Percentages.** Based on our results above, we employ Grad-K-Means as the primary grouping method, and trial incremental increases on the removal percentage for the retraining metric (using values of 1%, 5%, 10%, 20%). Our results in Figure 2 show that GGDA, in general, has a favorable accuracy efficiency tradeoff. In particular, we find that the performance levels are roughly equivalent across group sizes $\{1, 4, 16, 64\}$, but with orders-of-magnitude improvement in runtime.

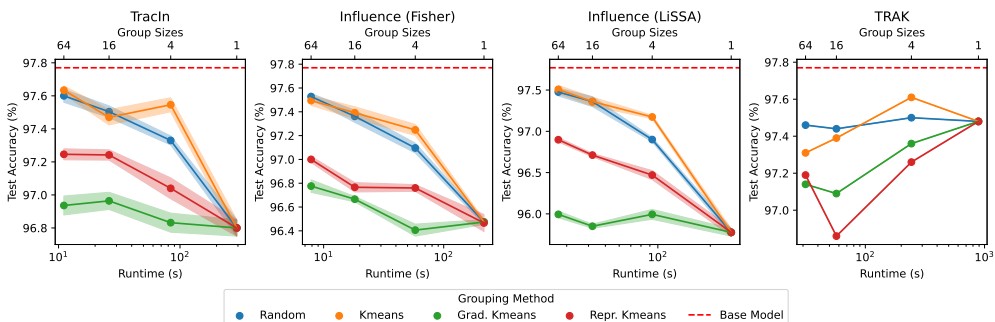

Figure 1: **Ablation over grouping methods.** Test accuracy (%) retraining score on MNIST after removing the top 20% most important training points (lower is better). Grad-K-Means grouping is superior, achieving comparable performance to individual attributions at orders of magnitudes faster runtime. Error bars represent standard error computed on 10 differently seeded model retrainings.

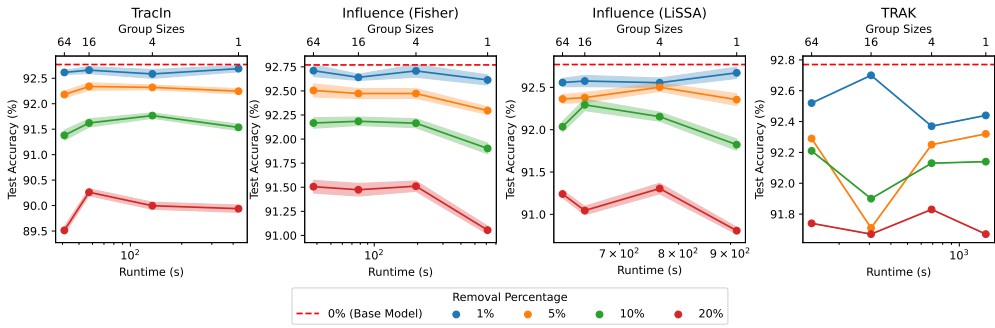

Figure 2: **GGDA attribution fidelity across removal percentages**. Test accuracy (%) retraining score using Grad-K-Means grouping on CIFAR-10 after removing 1%, 5%, 10%, and 20% of the most important points (lower is better). GGDA methods save orders of magnitude of runtime while gracefully trading off attribution fidelity. See Appendix C.1 for similar findings across all datasets and models.

### 5.4.2 DATASET PRUNING

To evaluate the effectiveness of GGDA in dataset pruning, we compute *test accuracy* after removing varying percentages (25%, 50%, 75%) of the least important points from the training set. We follow a similar sequential data removal as the retraining score evaluation. Higher test accuracies after pruning are more favorable, indicating that the removed points were indeed not influential for model performance. Our results demonstrate that GGDA significantly outperforms individual DA methods in this task, achieving higher test accuracies while requiring substantially less compute. Tables 1 and 2 present example results for a range of attribution methods compared between on the ANN dataset, illustrating the superior performance and efficiency of GGDA compared to individual DA methods in dataset pruning.

Table 1: Test accuracies (%) after pruning the HELOC dataset (LR) with varying removal percentages and for various DA/GGDA methods (using size 256 groupings). The full dataset (0% pruning) yielded 72.6% accuracy.

| Removal % | TracIn | | Inf. (Fisher) | | Inf. (LiSSA) | |
|---|---|---|---|---|---|---|
| | DA | GGDA | DA | GGDA | DA | GGDA |
| 25 | 70.8 ± 0.0 | 72.4 ± 0.0 | 57.0 ± 0.0 | 72.5 ± 0.0 | 70.8 ± 0.0 | 72.4 ± 0.0 |
| 50 | 56.3 ± 0.0 | 72.6 ± 0.0 | 50.7 ± 0.0 | 71.7 ± 0.0 | 56.5 ± 0.0 | 72.6 ± 0.0 |
| 75 | 47.9 ± 0.0 | 71.6 ± 0.0 | 42.5 ± 0.0 | 72.6 ± 0.0 | 47.9 ± 0.0 | 71.6 ± 0.0 |
| Runtime (s) | 6.18 ± 0.36 | 0.09 ± 0.03 | 6.15 ± 0.45 | 0.08 ± 0.00 | 10.7 ± 0.30 | 4.30 ± 0.04 |

Table 2: Test accuracies (%) after pruning the MNIST, CIFAR-10, and QNLI datasets with TracIn DA/GGDA methods and varying removal percentages (using size 1024 groupings).

| Removal % | MNIST | | CIFAR-10 | | QNLI | |
|---|---|---|---|---|---|---|
| | DA | GGDA | DA | GGDA | DA | GGDA |
| 0 | 97.7 ± 0.0 | | 92.6 ± 0.1 | | 85.4 ± 1.0 | |
| 25 | 95.8 ± 0.1 | 97.5 ± 0.0 | 89.9 ± 0.1 | 92.1 ± 0.1 | 74.9 ± 4.4 | 85.9 ± 1.2 |
| 50 | 93.6 ± 0.2 | 97.1 ± 0.1 | 84.4 ± 0.2 | 90.7 ± 0.1 | 64.8 ± 4.3 | 84.2 ± 1.6 |
| 75 | 85.8 ± 3.3 | 96.1 ± 0.1 | 74.4 ± 0.3 | 87.5 ± 0.1 | 46.4 ± 5.3 | 83.3 ± 3.5 |
| Runtime (s) | 311 ± 5.8 | 6.59 ± 0.07 | 492 ± 73.2 | 35.2 ± 0.07 | 1043 ± 14.3 | 100 ± 0.1 |

### 5.4.3 NOISY LABEL IDENTIFICATION

Noisy label detection is a critical task in machine learning, addressing the common issue of inaccurate labels in real-world datasets. These inaccuracies can arise from various sources, including

automated labeling processes, non-expert annotations, or intentional corruption by adversarial actors. Identifying and mitigating noisy labels is essential for maintaining model performance and reliability. In this context, we evaluate the effectiveness of GGDA in detecting noisy labels across different datasets and model architectures. Our analysis focuses on the trade-off between detection accuracy, measured by Area Under the Curve (AUC), and computational efficiency. We present our findings in Table 3, demonstrating superior performance for GGDA, particularly on image datasets.

Our analysis reveals several key findings regarding the performance of different DA methods in noisy label detection. For text classification using BERT models, the Fisher approximation variant of influence functions shows improved performance when combined with grouping, while also reducing runtime by an order of magnitude. TracIn demonstrates superior performance on the CIFAR-10 dataset, particularly when grouping datapoints with similar penultimate-layer gradients. However, we observe that TRAK performs poorly within the given computational budget, which can potentially be due to the number of ensembles being small (10 ensembles in our case). We choose not to further increase the ensemble as retraining make TRAK much slower compared to ours and all the baselines. Increasing it makes these DA methods not really comparable in terms of run time. Also, methods like Inf LiSSA show less pronounced group-wise speedups due to other computational bottlenecks. Nevertheless, applying GGDA to all of these DA methods improves their efficiency. These results highlight the effectiveness of GGDA in enhancing both performance and efficiency across various datasets and model architectures.

Table 3: Mean noisy label detection AUC and runtimes for increasingly large Grad-K-Means groups, measured across all datasets and attribution methods. For HELOC, we display results from the ANN-S model.

| Dataset | Attributor | Noisy Label Detection AUC / Runtime (s) per GGDA Group Size | | | | |
| | | 1 (DA) | 4 | 16 | 64 | 256 |
|---|---|---|---|---|---|---|
| HELOC | TracIn | 0.580 / 10.6 | 0.572 / 2.95 | 0.568 / 0.90 | 0.566 / 0.40 | 0.566 / 0.28 |
| | Inf. (Fisher) | 0.604 / 9.25 | 0.582 / 2.51 | 0.552 / 0.76 | 0.495 / 0.34 | 0.510 / 0.23 |
| | Inf. (LiSSA) | 0.650 / 18.5 | 0.631 / 11.9 | 0.627 / 10.2 | 0.621 / 9.77 | 0.620 / 9.65 |
| | TRAK | 0.504 / 46.4 | 0.549 / 12.0 | 0.528 / 3.92 | 0.512 / 1.22 | 0.503 / 0.77 |
| MNIST | TracIn | 0.637 / 208 | 0.646 / 55.9 | 0.650 / 18.1 | 0.642 / 8.18 | 0.529 / 5.69 |
| | Inf. (Fisher) | 0.509 / 287 | 0.551 / 90.7 | 0.586 / 27.8 | 0.554 / 9.49 | 0.502 / 4.58 |
| | Inf. (LiSSA) | 0.640 / 220 | 0.647 / 71.7 | 0.647 / 35.2 | 0.639 / 25.5 | 0.534 / 23.2 |
| | TRAK | 0.495 / 868 | 0.501 / 240 | 0.502 / 87.7 | 0.504 / 23.9 | 0.501 / 33.0 |
| CIFAR-10 | TracIn | 0.700 / 558 | 0.702 / 172 | 0.704 / 59.0 | 0.705 / 31.6 | 0.707 / 24.1 |
| | Inf. (Fisher) | 0.630 / 497 | 0.614 / 172 | 0.602 / 69.8 | 0.573 / 40.5 | 0.569 / 35.0 |
| | Inf. (LiSSA) | 0.656 / 835 | 0.657 / 544 | 0.661 / 468 | 0.678 / 452 | 0.685 / 447 |
| | TRAK | 0.501 / 1814 | 0.490 / 674 | 0.486 / 319 | 0.482 / 210 | 0.497 / 188 |
| QNLI | TracIn | 0.543 / 1743 | 0.539 / 427 | 0.537 / 172 | 0.536 / 136 | 0.536 / 119 |
| | Inf. (Fisher) | 0.667 / 2207 | 0.695 / 493 | 0.684 / 187 | 0.605 / 145 | 0.508 / 135 |
| | Inf. (LiSSA) | 0.501 / 2043 | 0.504 / 686 | 0.489 / 395 | 0.490 / 403 | 0.512 / 392 |
| | TRAK | 0.500 / 4947 | 0.498 / 1698 | 0.497 / 841 | 0.507 / 739 | 0.502 / 788 |

## 6 CONCLUSION

In this work, we have proposed a new framework, generalized group data attribution, that offers computational advantages over data attribution while maintaining similar fidelity. While our work has focussed more on influence-based approaches that are more non-trivial to extend to the group setting, any data attribution approach can, in principle, be extended to the group setting to obtain these computational benefits. The key insight here is that many downstream applications of attribution involve manipulating training datasets at scale, such as dataset pruning, or noisy label identification, and a fine-grained approach of individual point influence may be unnecessary in such cases.

In terms of future work, it is desirable to conduct a formal analysis of the fidelity tradeoffs in the group setting for various methods of interest, analytically quantifying the fidelity loss due to grouping.

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

APPENDIX

This appendix is formatted as follows:

- We provide full mathematical derivations in Appendix A.
- We provide experimental setup details in Appendix B.
    - Appendix B.1 contains dataset and model details.
    - Appendix B.2 contains clustering details.
    - Appendix B.3 contains attributor method parameters.
- Extended experimental results in full are located in Appendix C.

# A  DERIVATIONS

## A.1  GENERALIZED GROUP INFLUENCE FUNCTIONS

In this section, we present a derivation for group influence functions, mirroring the influence function derivation adopted by Koh & Liang (2017). The argument involves two steps. First, we analyse the change in model parameters upon upweighting the loss of a group of training points. Second, we compute how the change in model parameters affects the downstream property function. To begin, we consider what happens if we weight a group of training samples, $Z$. The modified loss and minimizer are:

$$L_\epsilon(\theta) = \frac{1}{n}\left[\sum_{i=1}^{n}\ell(z_i,\theta) + \epsilon\sum_{z\in Z}\ell(z,\theta)\right]$$

$$L_\epsilon(\theta) = L(\theta) + \frac{\epsilon}{n}\sum_{z\in Z}\ell(z,\theta) \text{ and } \hat{\theta}_\epsilon = \arg\min_\theta\left\{L(\theta) + \frac{\epsilon}{n}\sum_{z\in Z}\ell(z,\theta)\right\}$$

$$\text{With gradient } \nabla L_\epsilon(\theta) = \nabla L(\theta) + \frac{\epsilon}{n}\sum_{z\in Z}\nabla\ell(z,\theta)$$

Let $\partial\theta = (\hat{\theta}_\epsilon - \hat{\theta})$. For a single step:

$$f(\hat{\theta}_\epsilon) \approx f(\hat{\theta}) + \nabla f(\hat{\theta})\partial\theta$$

We now make the following simplifying assumptions, following Koh & Liang (2017)

**Assumption 1:** For small $\epsilon$, the modified loss $L_\epsilon$ is locally linear between $\hat{\theta}$ and $\hat{\theta}_\epsilon$.

We can approximate the gradient $\nabla L_\epsilon(\hat{\theta}_\epsilon)$ by starting from $\hat{\theta}$ and taking a single step towards $\hat{\theta}_\epsilon$ (for $f = \nabla L_\epsilon$):

$$\nabla L_\epsilon(\hat{\theta}_\epsilon) \approx \nabla L_\epsilon(\hat{\theta}) + \nabla^2 L_\epsilon(\hat{\theta})\partial\theta$$

**Assumption 2:** $\hat{\theta}_\epsilon$ minimizes $L_\epsilon \implies \nabla L_\epsilon(\hat{\theta}_\epsilon) = 0$

$$\nabla L_\epsilon(\hat{\theta}) + \nabla^2 L_\epsilon(\hat{\theta})\partial\theta = 0$$

$$\implies [\nabla L(\hat{\theta}) + \frac{\epsilon}{n}\sum_{z\in Z}\nabla\ell(z,\hat{\theta})] + [\nabla^2 L(\hat{\theta}) + \frac{\epsilon}{n}\sum_{z\in Z}\nabla^2\ell(z,\hat{\theta})]\partial\theta = 0$$

Rearranging, we have

$$\partial\theta = -[\nabla^2 L(\hat{\theta}) + \frac{\epsilon}{n}\sum_{z\in Z}\nabla^2\ell(z,\hat{\theta})]^{-1}[\nabla L(\hat{\theta}) + \frac{\epsilon}{n}\sum_{z\in Z}\nabla\ell(z,\hat{\theta})]$$

Using $(A + \epsilon B)^{-1} \approx A^{-1} - \epsilon A^{-1} B A^{-1}$, we set $A = \nabla^2 L(\hat{\theta})$ and $B = \frac{1}{n} \sum_{z \in Z} \nabla^2 \ell(z, \hat{\theta})$, yielding:

$$\partial \theta = - \underbrace{\left[ [\nabla^2 L(\hat{\theta})]^{-1} - \frac{\epsilon}{n} \sum_{z \in Z} [\nabla^2 L(\hat{\theta})]^{-1} \nabla^2 \ell(z, \hat{\theta}) [\nabla^2 L(\hat{\theta})]^{-1} \right]}_{(A+\epsilon B)^{-1} = [\nabla^2 L(\hat{\theta}) + \frac{\epsilon}{n} \sum_{z \in Z} \nabla^2 \ell(z, \hat{\theta})]^{-1}} \left[ \nabla L(\hat{\theta}) + \frac{\epsilon}{n} \sum_{z \in Z} \nabla \ell(z, \hat{\theta}) \right]$$

**Assumption 3:** $\hat{\theta}$ minimizes $L \implies \nabla L(\hat{\theta}) = 0$

$$\partial \theta = - \left[ [\nabla^2 L(\hat{\theta})]^{-1} - \frac{\epsilon}{n} \sum_{z \in Z} [\nabla^2 L(\hat{\theta})]^{-1} \nabla^2 \ell(z, \hat{\theta}) [\nabla^2 L(\hat{\theta})]^{-1} \right] \frac{\epsilon}{n} \sum_{z \in Z} \nabla \ell(z, \hat{\theta})$$

As $\epsilon \to 0$, we have $\epsilon = \partial \epsilon$ and drop $o(\epsilon)$ terms:

$$\partial \theta = -\nabla^2 L(\hat{\theta})^{-1} \frac{\epsilon}{n} \sum_{z \in Z} \nabla \ell(z, \hat{\theta})$$

$$\implies \frac{\partial \theta}{\partial \epsilon} = -\nabla^2 L(\hat{\theta})^{-1} \frac{1}{n} \sum_{z \in Z} \nabla \ell(z, \hat{\theta})$$

**Assumption 4:** Hessian at $\theta = \hat{\theta}$ exists and is positive definite.

$$\implies \frac{\partial \theta}{\partial \epsilon} = -\frac{1}{n} H_{\hat{\theta}}^{-1} \sum_{z \in Z} \nabla \ell(z, \hat{\theta})$$

This is the rate of change of parameters $\theta$ as we uniformly upweight a group of training samples $Z$ by $\epsilon$ in the loss function (note that upweighting occurs before normalizing by $1/n$). This concludes the first part of the analysis, i.e., computing the change in model parameters due to the up-weighting.

The second is fairly trivial, as the gradient of the property function gives the change in property function with the change in model parameters. Combining both, the overall attribution of a group to some test property, $g(\theta)$ is:

$$\frac{\partial g}{\partial \epsilon} = \frac{\partial g}{\partial \theta} \times \frac{\partial \theta}{\partial \epsilon} = -\frac{1}{n} \nabla g H_{\hat{\theta}}^{-1} \sum_{z \in Z} \nabla \ell(z, \hat{\theta})$$

## A.2 TRACIN AS A SIMPLIFIED INFLUENCE APPROXIMATION

Pruthi et al. (2020) propose an influence definition based on aggregating the influence of model checkpoints in gradient descent. Formally, tracein can be described as follows:

$$\tau_{\text{identity-inf}}(\theta, \mathcal{D}, \mathbf{x}_{\text{test}}) = \nabla_\theta \ell(\mathbf{x}_{\text{test}}; \theta)^\top \left[ \nabla_\theta \ell(\mathbf{x}_0; \theta); \nabla_\theta \ell(\mathbf{x}_1; \theta); ... \nabla_\theta \ell(\mathbf{x}_n; \theta) \right] \quad (4)$$

$$\tau_{\text{tracein}}(\mathcal{A}, \mathcal{D}, \mathbf{x}_{\text{test}}) = \sum_{i=1}^{T} \tau_{\text{identity-inf}}(\theta_i, \mathcal{D}, \mathbf{x}_{\text{test}}) \quad \text{where } \{\theta_i | i \in [1, T]\} \text{ are checkpoints} \quad (5)$$

The core DA method used by tracein is the standard influence functions, with the Hessian being set to identity, which drastically reduces computational cost. The GGDA extension of tracein thus amounts to replacing the `identity-inf` component (equation 4) with equation 1, with the Hessian set to identity.

### A.3 Background on Fisher information matrix

One of the popular techniques for Hessian approximation in machine learning is via the Fisher Information Matrix. When the quantity whose Hessian is considered is a log-likelihood term, it turns out that the Hessian has a simple form:

$$H_\theta = -\sum_{\mathbf{x}} \mathbb{E}_{y' \sim p(y|x)} \nabla^2_\theta \log p(y' \mid \mathbf{x}, \theta)$$

$$= \sum_{\mathbf{x}} \mathbb{E}_{y' \sim p(y|x)} \nabla_\theta \log p(y' \mid \mathbf{x}, \theta) \nabla_\theta \log p(y' \mid \mathbf{x}, \theta)^\top = F_\theta$$

Here, the term $F_\theta$ is the Fisher Information Matrix. Thus, when the loss term is a log-likelihood term, $\ell(\mathbf{x}, \theta) = \mathbb{E}_{y' \sim p(y|x;\theta)} \log p(y' \mid \mathbf{x}; \theta)$, the Hessian can be written in this manner (Barshan et al., 2020). In practical applications involving the Fisher, it is common to use the so-called Empirical Fisher matrix, which is given by:

$$\hat{F}_\theta = \sum_{\mathbf{x}} \nabla_\theta \log p(\mathbf{y} \mid \mathbf{x}, \theta) \nabla_\theta \log p(\mathbf{y} \mid \mathbf{x}, \theta)^\top \quad \text{(where } \mathbf{y} \text{ is the ground truth label for } \mathbf{x}) \quad (6)$$

While the limitations of using the empirical Fisher approximation have been well-documented (Kunstner et al., 2019), primarily owing to issues with the bias of the estimate, it is employed practically as a convenient approximation. Computationally, while the Hessian of log-likelihood requires both a double-backpropagation and a Monte Carlo expectation, the Fisher only requires a single backpropagation for gradient computation with the Monte Carlo estimate. On the other hand, the empirical Fisher approximation forgoes the Monte Carlo estimate.

### A.4 TRAK as Ensembled Fisher Influence

TRAK Park et al. (2023) is an influence method derived for binary classification setting. Specifically, given a logistic classifier $\ell(\theta) = \log(1 + \exp(-y_{gt} f(\mathbf{x})))$, TRAK without projection + ensembling (Equation 13 in the TRAK paper) is equivalent to the following (note distinction between $\ell$ vs $f$):

$$\tau_{\text{trak}} = \underbrace{\nabla_\theta f(\mathbf{x}_{test}; \theta)}_{\mathbb{R}^{1 \times p}} \underbrace{\hat{H}_\theta^{-1}}_{\mathbb{R}^{p \times p}} \underbrace{\left[ \nabla_\theta \ell(\mathbf{x}_0; \theta); \nabla_\theta \ell(\mathbf{x}_1; \theta); ... \nabla_\theta \ell(\mathbf{x}_n; \theta) \right]}_{\mathbb{R}^{p \times n}}$$

$$\hat{H}_\theta = \sum_{i=1}^n \nabla_\theta f(\mathbf{x}_i; \theta) \nabla_\theta f(\mathbf{x}_i; \theta)^\top$$

Comparing the Hessian approximation term $\hat{H}$ with the formula for the empirical Fisher approximation in the main paper, we observe that TRAK computes outer products of model outputs, whereas the empirical Fisher involves outer products of the loss function. We show below that these two quantities are related:

$$\ell(\mathbf{x}, \theta) = \log(1 + \exp(-y_{gt} f(\mathbf{x}, \theta)/T))$$

$$\nabla_\theta \ell(\mathbf{x}, \theta) = \frac{\exp(-y_{gt} f(\mathbf{x}, \theta)/T)}{1 + \exp(-y_{gt} f(\mathbf{x}, \theta)/T)} (-y_{gt}/T) \nabla_\theta f(\mathbf{x}, \theta)$$

$$\nabla_\theta \ell(\mathbf{x}_i; \theta) \nabla_\theta \ell(\mathbf{x}_i; \theta)^\top = \underbrace{\frac{\exp(-2y_{gt} f(\mathbf{x}, \theta)/T)}{(1 + \exp(-y_{gt} f(\mathbf{x}, \theta)/T))^2 T^2}}_{C(T)} \nabla_\theta f(\mathbf{x}_i; \theta) \nabla_\theta f(\mathbf{x}_i; \theta)^\top \quad (y_{gt} \in \{+1, -1\})$$

Where $T$ is a temperature parameter. If we set $T \to \infty$, then $C(T) \to \frac{1}{4T^2}$. Thus in this case,

$$\nabla_\theta \ell(\mathbf{x}_i; \theta) \nabla_\theta \ell(\mathbf{x}_i; \theta)^\top \propto \nabla_\theta f(\mathbf{x}_i; \theta) \nabla_\theta f(\mathbf{x}_i; \theta)^\top$$

Note that the temperature is a post-hoc scaling parameter that can be added after training. It is thus unrelated to the underlying learning algorithm, as it simply scales the confidence or loss values. Therefore, the single model influence estimates in TRAK is equivalent to Influence functions with an empirical Fisher approximation.

## B    EXPERIMENTAL SETUP

### B.1    DATASETS AND MODELS

Here we provide further details regarding the specifics of the datasets and models used.

#### B.1.1    DATASET DETAILS

**HELOC** (Home Equity Line of Credit)   A financial dataset used for credit risk assessment. It contains 9,871 instances with 23 features, representing real-world credit applications. We use 80% (7,896 instances) for training and 20% (1,975 instances) for testing. The task is a binary classification problem to predict whether an applicant will default on their credit line. For this dataset, we employ three models: LR, and two ANNs with ReLU activation functions. ANN-S has two hidden layers both with size 50, while ANN-M has two hidden layers with sizes 200 and 50.

**MNIST**   A widely-used image dataset of handwritten digits for image classification tasks (LeCun et al., 1998). We use the standard train/test split of 60,000/10,000 images. For MNIST, we utilize an ANN with ReLU activation and three hidden layers, each with 150 units.

**CIFAR-10**   A more complex image classification dataset consisting of 32x32 color images in 10 classes (Krizhevsky & Hinton, 2009). We use the standard train/test split of 50,000/10,000 images. For CIFAR-10, we employ ResNet-18 (He et al., 2016).

**QNLI** (Question-answering Natural Language Inference)   A text classification dataset derived from the Stanford Question Answering Dataset (Wang & Manning, 2018). It contains 108,436 question-sentence pairs, with the task of determining whether the sentence contains the question's answer. Following TRAK (Park et al., 2023), we sample 50,000 instances for train and test on the standard validation set of 5,463 instances. For QNLI, we utilize a pre-trained BERT (Devlin, 2018) model with an added classification head, obtained from Hugging Face's Transformers library (Wolf, 2019).

#### B.1.2    MODEL HYPERPARAMETERS

Across all datasets and models, we use a (mean-reduction) cross entropy loss function with Adam optimizer on PyTorch's default OneCycleLR scheduler (besides for QNLI/BERT, where a linear scheduler is used). Weight decay is utilized in the case of training on tabular data to prevent overfitting.

Table 4: Hyperparameters for different models and datasets.

| Dataset | Model | Scheduler | Learning Rate | Epochs | Batch Size | Weight Decay |
|---------|-------|-----------|---------------|--------|------------|--------------|
| HELOC | LR | OneCycle | 0.1 | 200 | 128 | 0 |
| HELOC | ANN-S | OneCycle | 0.001 | 300 | 128 | 4e-4 |
| HELOC | ANN-M | OneCycle | 5e-4 | 400 | 128 | 4e-4 |
| MNIST | ANN-L | OneCycle | 0.001 | 50 | 512 | 0 |
| CIFAR-10 | ResNet18 | OneCycle | 0.01 | 50 | 512 | 0 |
| QNLI | BERT | Linear | 2e-5 | 3 | 512 | 0 |

### B.2    GROUPING HYPERPARAMETERS

We whiten all inputs to KMeans clustering (raw inputs, intermediate representations, and penultimate-layer gradients). Batch sizes for distance computations are computed programmati-

cally based on available GPU memory. We use a convergence tolerance of 0.001 on the center shift of the KMeans algorithm, with the maximum number of iterations set to 60.

### B.3    BASELINE ATTRIBUTORS

We implement TracIn using the final model checkpoint. Random projection dimensions used in Inf. Fisher and TRAK are 16, 32, 64, and 32 for HELOC, MNIST, CIFAR-10 and QNLI respectively, owing to GPU constraints in the case of QNLI. TRAK follows Park et al. (2023) with a subsampling fraction of 0.5. Inf. LiSSA contains the largest number of parameters to choose from, and we use recommended values of damp = 0.001, repeat = 20, depth = 200, scale = 50.

## C    ADDITIONAL RESULTS

This appendix details remaining results from our extensive experiments across all datasets, models, attribution methods, grouping methods, and group sizes.

### C.1    RETRAINING SCORES

Ablations over grouping methods are displayed for all HELOC models in Figure 3, CIFAR-10 / ResNet18 in Figure 4, and QNLI / BERT in Figure 5. While random grouping necessarily provides orders of magnitude speedups in runtime, it is inferior at grouping together points that similarly impact model performance. For instance, the most important Random groups of size 64 consisted of a rough 50:50 split between points that had positive and negative individual attributions, while the most important Grad-K-Means group of size 64 consisted only of points with positive individual attributions (across all three attribution methods shown). In terms of preserving group integrity, the Grad-K-Means method outperforms other approaches by consistently grouping together data points that influence model performance in similar ways.

Removal percentage sweeps are also included for HELOC in Figure 6, MNIST in Figure 7, and QNLI in Figure 8. Overall, we observe that GGDA methods save orders of magnitude of runtime while gracefully trading off attribution fidelity.

### C.2    DATASET PRUNING

Extended results for the dataset pruning experiments described in the main text can be found in Tables 5 through 9. As in the main results, GGDA methods yield similar or better dataset pruning efficacy while drastically reducing runtime.

Table 5: Test accuracies (%) after pruning the HELOC dataset (ANN-S) with varying removal percentages and for various DA/GGDA methods (using size 256 groupings). The full dataset (0% pruning) yielded 73.5% ± 0.2 accuracy.

| Removal % | TracIn | | Inf. (Fisher) | | Inf. (LiSSA) | |
|---|---|---|---|---|---|---|
| | DA | GGDA | DA | GGDA | DA | GGDA |
| 25 | 70.7 ± 0.1 | 73.9 ± 0.3 | 68.0 ± 0.8 | 73.9 ± 0.2 | 68.5 ± 0.2 | 73.8 ± 0.3 |
| 50 | 53.6 ± 0.0 | 72.8 ± 0.4 | 54.2 ± 0.7 | 73.2 ± 0.4 | 52.1 ± 0.0 | 72.8 ± 0.4 |
| 75 | 47.9 ± 0.0 | 71.4 ± 0.6 | 45.3 ± 0.8 | 71.9 ± 0.8 | 52.3 ± 0.2 | 70.8 ± 0.8 |
| Runtime (s) | 7.81 ± 0.53 | 0.25 ± 0.05 | 7.33 ± 0.67 | 0.24 ± 0.04 | 14.8 ± 0.64 | 8.06 ± 0.05 |

### C.3    NOISY LABEL DETECTION

Results of the noisy label detection task for the remaining datasets and models are provided in Table 10.

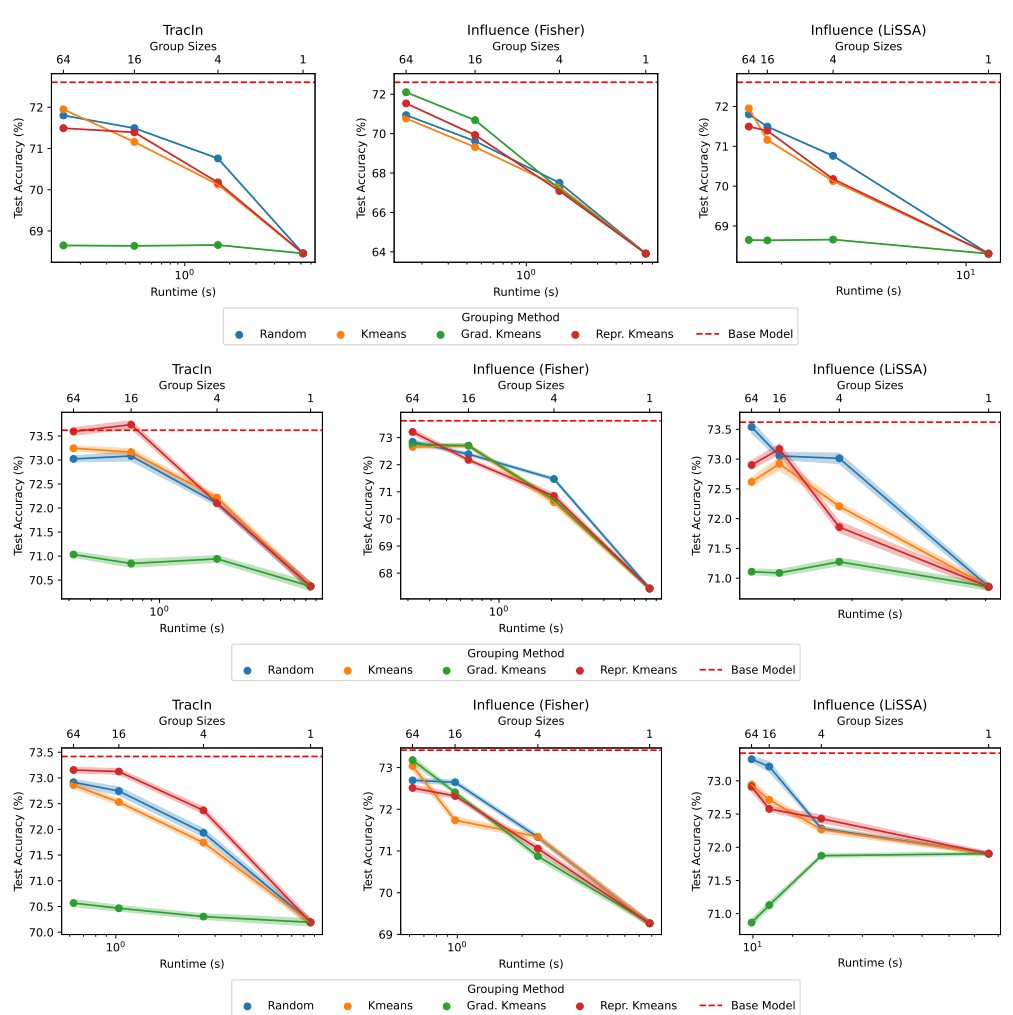

Figure 3: **Ablation over grouping methods.** Top: LR. Middle: ANN-S. Bottom: ANN-M. Test accuracy (%) retraining score on HELOC after removing the top 20% most important training points (lower is better).

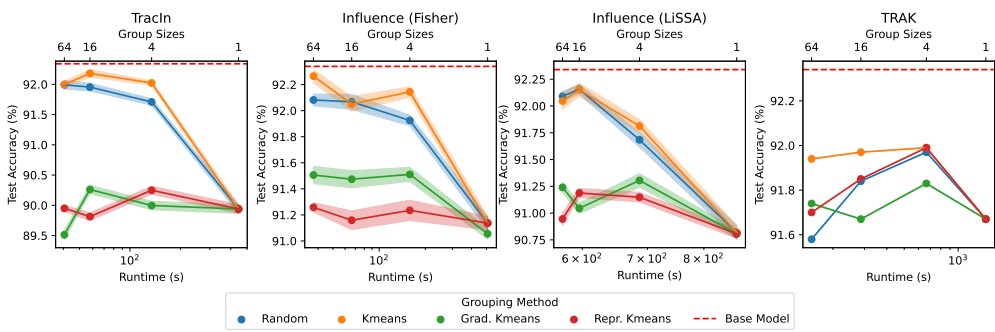

Figure 4: **Ablation over grouping methods.** Test accuracy (%) retraining score on CIFAR-10 after removing the top 20% most important training points (lower is better).

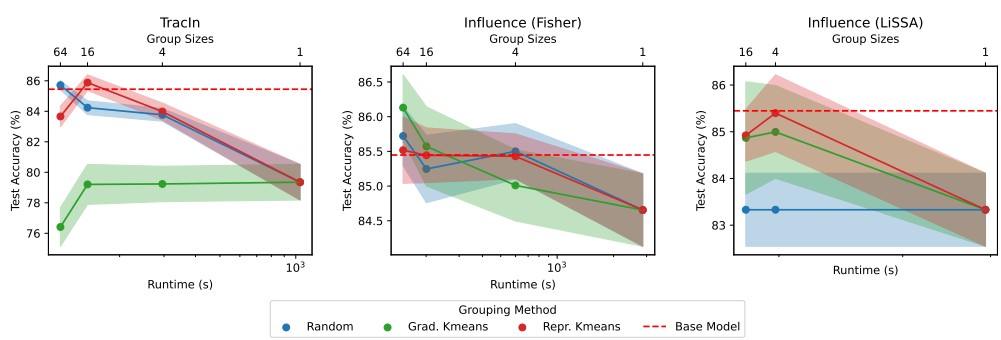

Figure 5: **Ablation over grouping methods.** Test accuracy (%) retraining score on QNLI after removing the top 20% most important training points (lower is better).

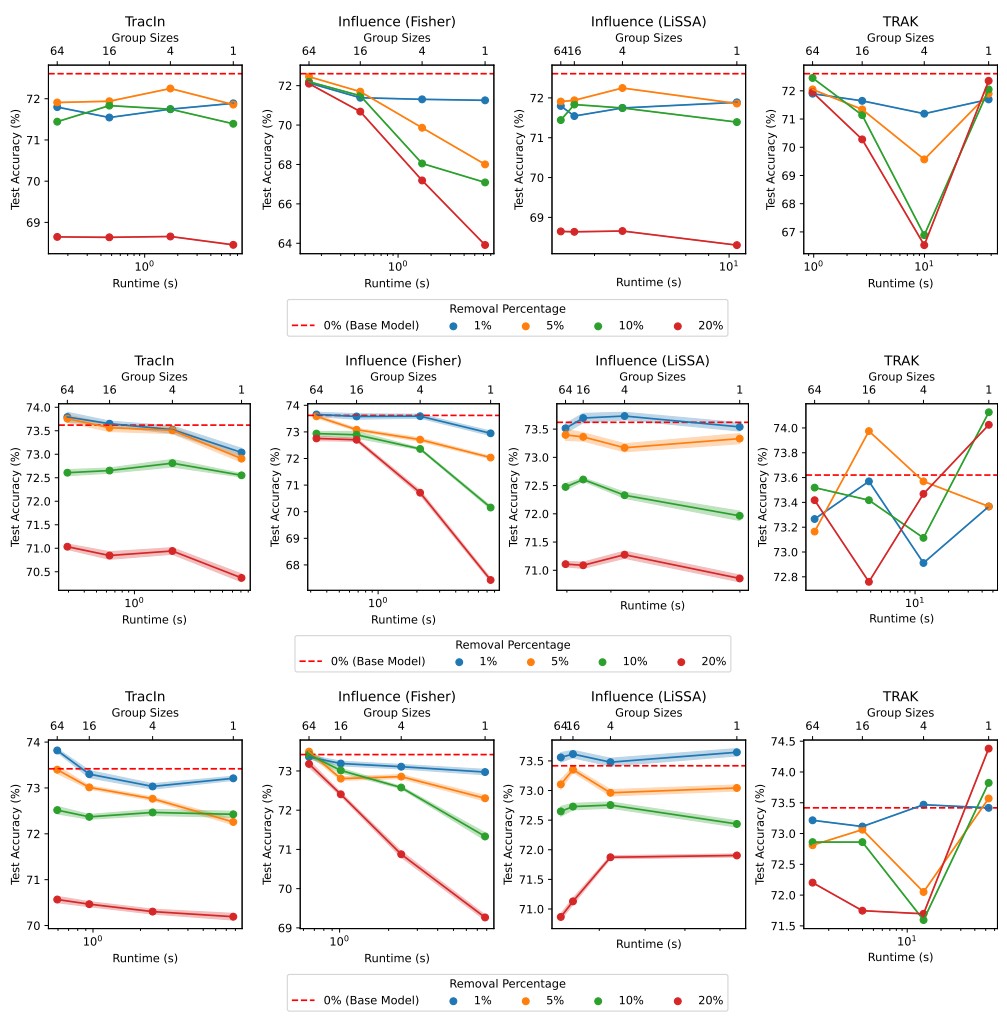

Figure 6: **GGDA attribution fidelity across removal percentages**. Top: LR. Middle: ANN-S. Bottom: ANN-M. Test accuracy (%) retraining score using Grad-K-Means grouping on HELOC after removing 1%, 5%, 10%, and 20% of the most important points (lower is better).

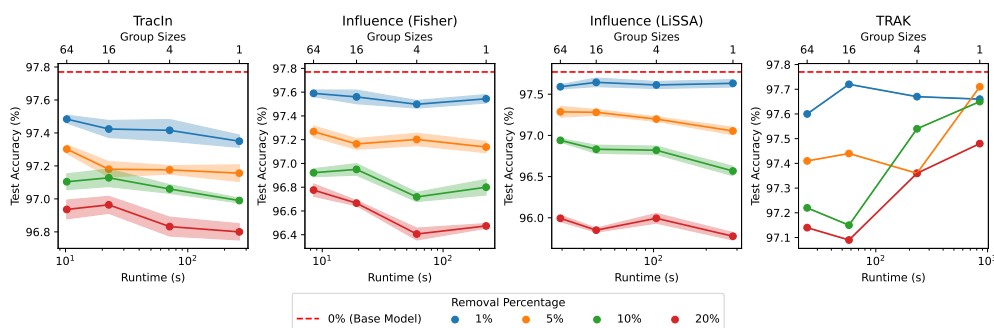

Figure 7: **GGDA attribution fidelity across removal percentages**. Test accuracy (%) retraining score using Grad-K-Means grouping on MNIST after removing 1%, 5%, 10%, and 20% of the most important points (lower is better).

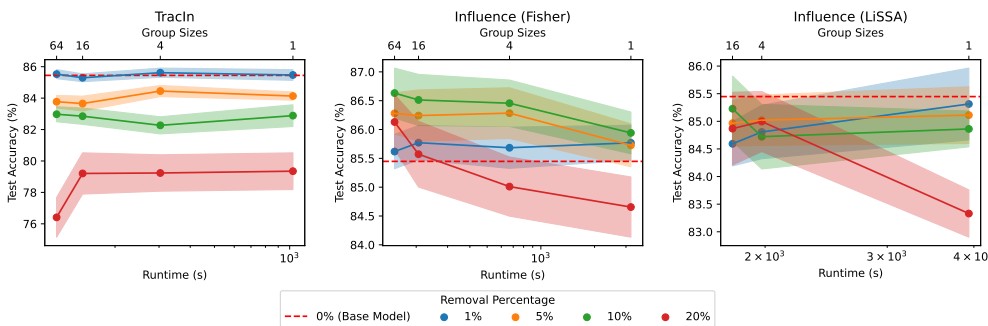

Figure 8: **GGDA attribution fidelity across removal percentages**. Test accuracy (%) retraining score using Grad-K-Means grouping on QNLI after removing 1%, 5%, 10%, and 20% of the most important points (lower is better).

Table 6: Test accuracies (%) after pruning the HELOC dataset (ANN-M) with varying removal percentages and for various DA/GGDA methods (using size 256 groupings). The full dataset (0% pruning) yielded 73.4% ± 0.1 accuracy.

| | **TracIn** | | **Inf. (Fisher)** | | **Inf. (LiSSA)** | |
|---|---|---|---|---|---|---|
| Removal % | DA | GGDA | DA | GGDA | DA | GGDA |
| 25 | 71.2 ± 0.2 | 73.3 ± 0.2 | 71.0 ± 0.4 | 73.7 ± 0.3 | 68.7 ± 0.2 | 72.6 ± 0.2 |
| 50 | 52.9 ± 0.1 | 72.5 ± 0.4 | 56.1 ± 0.4 | 72.4 ± 0.2 | 51.2 ± 0.0 | 71.4 ± 0.2 |
| 75 | 47.9 ± 0.0 | 68.0 ± 0.5 | 49.3 ± 0.3 | 69.7 ± 0.6 | 48.3 ± 0.1 | 70.3 ± 0.7 |
| Runtime (s) | 9.51 ± 0.03 | 0.52 ± 0.03 | 7.77 ± 0.48 | 0.54 ± 0.05 | 16.5 ± 0.67 | 9.72 ± 0.08 |

Table 7: Test accuracies (%) after pruning the MNIST dataset with varying removal percentages and for various DA/GGDA methods (using size 1024 groupings). The full dataset (0% pruning) yielded 97.7% ± 0.1 accuracy.

| | **TracIn** | | **Inf. (Fisher)** | | **Inf. (LiSSA)** | |
|---|---|---|---|---|---|---|
| Removal % | DA | GGDA | DA | GGDA | DA | GGDA |
| 25 | 95.8 ± 0.1 | 97.5 ± 0.0 | 96.7 ± 0.1 | 97.4 ± 0.0 | 96.0 ± 0.1 | 97.4 ± 0.1 |
| 50 | 93.6 ± 0.2 | 97.1 ± 0.1 | 96.3 ± 0.1 | 97.1 ± 0.1 | 94.0 ± 0.3 | 97.0 ± 0.1 |
| 75 | 85.8 ± 3.3 | 96.1 ± 0.1 | 95.6 ± 0.1 | 95.9 ± 0.1 | 92.6 ± 0.6 | 95.9 ± 0.1 |
| Runtime (s) | 311 ± 5.8 | 6.59 ± 0.07 | 227 ± 3.3 | 5.09 ± 0.04 | 307 ± 1.7 | 25.9 ± 0.70 |

Table 8: Test accuracies (%) after pruning the CIFAR-10 dataset with varying removal percentages and for various DA/GGDA methods (using size 1024 groupings). The full dataset (0% pruning) yielded 92.7% ± 0.1 accuracy.

| Removal % | TracIn DA | TracIn GGDA | Inf. (Fisher) DA | Inf. (Fisher) GGDA | Inf. (LiSSA) DA | Inf. (LiSSA) GGDA |
|---|---|---|---|---|---|---|
| 25 | 89.9 ± 0.1 | 92.1 ± 0.1 | 90.9 ± 0.2 | 92.1 ± 0.1 | 90.1 ± 0.1 | 92.0 ± 0.2 |
| 50 | 84.4 ± 0.2 | 90.7 ± 0.1 | 89.0 ± 0.1 | 90.6 ± 0.2 | 88.1 ± 0.3 | 90.7 ± 0.1 |
| 75 | 74.4 ± 0.3 | 87.5 ± 0.1 | 86.9 ± 0.2 | 87.6 ± 0.3 | 83.4 ± 0.3 | 87.5 ± 0.2 |
| Runtime (s) | 493 ± 73 | 35.2 ± 0.07 | 469 ± 41.3 | 33.6 ± 0.15 | 827 ± 4.1 | 443 ± 0.61 |

Table 9: Test accuracies (%) after pruning the QNLI dataset with varying removal percentages and for various DA/GGDA methods (using size 1024 groupings). The full dataset (0% pruning) yielded 85.4% ± 1.0 accuracy.

| Removal % | TracIn DA | TracIn GGDA | Inf. (Fisher) DA | Inf. (Fisher) GGDA |
|---|---|---|---|---|
| 25 | 74.9 ± 4.4 | 85.9 ± 1.2 | 85.2 ± 1.5 | 85.0 ± 1.7 |
| 50 | 64.8 ± 4.3 | 84.2 ± 1.6 | 83.3 ± 1.7 | 84.7 ± 1.2 |
| 75 | 46.4 ± 5.3 | 83.3 ± 3.5 | 75.2 ± 4.0 | 82.7 ± 1.5 |
| Runtime (s) | 1043 ± 14.3 | 100.2 ± 0.11 | 2191 ± 48.4 | 114.9 ± 0.83 |

Table 10: Mean noisy label detection AUC and runtimes for increasingly large Grad-K-Means groups (remaining datasets and models).

| Dataset | Attributor | Noisy Label Detection AUC / Runtime (s) per GGDA Group Size 1 (DA) | 4 | 16 | 64 | 256 |
|---|---|---|---|---|---|---|
| HELOC / LR | TracIn | 0.555 / 7.44 | 0.545 / 1.96 | 0.539 / 0.54 | 0.541 / 0.18 | 0.539 / 0.08 |
| | Inf. (Fisher) | 0.614 / 7.43 | 0.581 / 1.95 | 0.558 / 0.52 | 0.520 / 0.17 | 0.512 / 0.08 |
| | Inf. (LiSSA) | 0.575 / 13.54 | 0.572 / 7.38 | 0.571 / 5.78 | 0.572 / 5.39 | 0.573 / 5.30 |
| HELOC / ANN-M | TracIn | 0.597 / 11.44 | 0.560 / 3.44 | 0.553 / 1.29 | 0.549 / 0.75 | 0.550 / 0.60 |
| | Inf. (Fisher) | 0.597 / 13.39 | 0.571 / 4.05 | 0.563 / 1.47 | 0.536 / 0.81 | 0.533 / 0.63 |
| | Inf. (LiSSA) | 0.636 / 24.10 | 0.601 / 16.70 | 0.595 / 14.43 | 0.592 / 13.75 | 0.592 / 13.70 |

