# OpenReview forum: "Generalized Group Data Attribution"
_ICLR.cc/2025/Conference — Submitted to ICLR 2025_

### Official Review · Reviewer_cACW · 2024-10-24

**Soundness:** 3
**Presentation:** 3
**Contribution:** 2
**Rating:** 5
**Confidence:** 4

**Summary:**

This paper proposes to generalize the traditional data attribution method to consider attributing to (1) groups of data points instead of individual data points, and (2) the property function that captures model behavior beyond the test loss.

**Strengths:**

Overall
1. The motivation is clear.
2. The definitions and the problem formulation are consistent throughout.
3. The experimental results are presented cleanly.

**Weaknesses:**

The main weaknesses are the novelty and soundness of both proposals:
1. The second proposal, "property function," is already known in the literature as "target function." For example, see [1] in statistics, and [2] in data attribution.
2. The first proposal, considering groups of data points, doesn't seem to have the claimed computational gain by looking at the analysis for the Influence Function.
    - From Section 4, the analysis is true obviously since there are no conceptual differences from the influence function (together with the target function concept mentioned above). While I'm not sure about the computational complexity advantage claimed in the paper (batch gradient computation $\\approx$ per gradient computation), from my understanding, for whatever algorithm is used to approximate the iHVP (inverse-Hessian vector product) computation, constructing $H_{\\theta} = \\sum_{i=1}^{n} \\nabla_\\theta^2 \\ell(x_i)$ will inevitably scale with $n$. Hence, the claimed computational advantage is not there at least until Line 266.
    - It's better to bring the idea of batched $H_{\\theta}$ as in the TRAK paragraph for $\\hat{F}_{\\theta}^{\\text{batched}}$ to the Influence Function analysis to demonstrate the claimed advantage.

       However, even with the batched approximation of $H_{\\theta}$, some theoretical justification is lacking. Whether this will be a good approximation is unclear to me. Additionally, as described in Line 319, some clustering algorithms are needed in order to obtain a small approximation error on $\\nabla_{\\theta}\\ell(x_i)$, which I suppose will make the algorithm scale with $n$ again.

Overall, the second proposal (*property function*) already exists in the literature, while for the first proposal (*grouped data points*), I'm not convinced by the claimed computational efficiency gain for the Influence function and TRAK when Hessian or (empirical) FIM are involved. Without a justification for an efficient and good approximation of the Hessian (and its inverse with iHVP computation), such an extension is trivial from the linearity of IF, TRAK, and related influence-function-based methods.

**Reference**
- [1]: [An Automatic Finite-Sample Robustness Metric: When Can Dropping a Little Data Make a Big Difference?](http://arxiv.org/abs/2011.14999)
- [2]: [Most Influential Subset Selection: Challenges, Promises, and Beyond](https://arxiv.org/abs/2409.18153)

**Questions:**

See Weaknesses. Additionally:
1. In Section 4, the definition of TracIn is not standard, at least deviating from the original paper and even ignoring the scaling. It only sums over batches that contain a particular training sample $x_i$, not over all iterations unless we're considering full-batch training. I think this should be mentioned.
2. The *Gradient K-Means* grouping method in the experiment suffers from the issue I raise above (2), where when we need to consider individual gradients, the claimed computational efficiency goes away.

Some minor suggestions in writing:
1. Line 120, replace $\mathcal{D} = \\{(x_0, y_0), (x_1, y_1), \\ldots (x_n y_n)\\}$ by $\\mathcal{D} = \\{(x_0, y_0), (x_1, y_1), \\ldots , (x_n, y_n)\\}$ (two "," are missing).
2. Line 217 and 266, replace $k << n$ by $k \\ll n$.
3. Line 232, replace - with --- without spaces at the beginning (before TRAK) and the ending (after TracIn).
4. Line 252, replace `\citep{}` with `\citet{}`.
5. Line 276~288, tracein should all be TracIn?

---

> ### Author Response · Authors · 2024-11-22
> **Rebuttal by Authors**
>
> Thank you for your thoughtful review of our paper. We would like to address the weaknesses you pointed out and provide clarification on the questions raised.
>
> > W1: Property function is not a contribution
>
> We agree that the property function is not a novel contribution of our work. We simply wanted to highlight that our method is capable of handling general property functions as an additional feature. We have changed the writing in our revised version to clarify this point and avoid any potential misunderstanding about our claimed contributions.
>
>
> > W2a: Computational advantage doesn’t apply to Hessian computation
>
> This is a very astute observation regarding the Hessian! It is true that modifying the Hessian estimation is critical to reducing the computational complexity of attribution estimation. While our draft has neglected to mention this in detail, our implementation also subsequently involves Hessian approximations, where we denote the Hessian as $$H_{\theta} = \sum_z \nabla^2 \ell(z)$$, thus rendering it to be a sum over number of groups. We mention this for TRAK, where we refer to the subsequent Fisher approximation as a “batched” Fisher approximation. We perform similar “batched” Hessian approximations for influence functions using the LiSSA estimator to ensure that per-sample computations are never performed. We again apologize for neglecting to discuss these batched Hessian approximations in our draft!
>
>
> > W2b:  Whether batched approximation is good
>
> We agree with the reviewer’s intuition that clustering algorithms are required to ensure a small approximation error. We emphasize that this is exactly the purpose of our group selection strategies! In particular, we found the gradient-based clustering to be highly effective, pointing to exactly the approximation benefit described by the reviewer.
>
>
> > Q1: TracIn definition is not standard
>
> Thanks for bringing this to our notice! Our definition in equation (3) does deviate from the one proposed in the TracIn paper with a constant term involving the learning rate pre-multiplier. If a constant learning rate schedule is used throughout training, our variant is equivalent to the one proposed in the TracIn paper.
>
> Please see the discussion in Appendix A.2 and A.3 for a complete discussion of the TracIn variant we employ.
>
> > Q2: Grad-K-Means requires individual gradients
>
> We agree that a full gradient computation will brings down efficiency. Therefore, we choose to use gradients wrt last layer activations and NOT gradients wrt parameters as we discuss in line 354. Please also note that, as discussed in the paper, it is hard to efficiently compute batched versions of gradients wrt parameters, but not so for gradients wrt activations, whose batched versions can be trivially computed. While we agree that any grouping method must scale with $n$, our grouping method in practice is cheap and does not involve computing per-sample gradients wrt weights. Furthermore, these groups only need to be computed once offline, and do not need to be recomputed for subsequent influence estimations for new test data points / model properties.
>
> > Q3: Typos
>
> Thank you for pointing out the typos. We have fixed them in the revised version of the paper.
>
> > Remark
>
> Thank you once again for your insightful suggestions and comments, which have been instrumental in enhancing the quality of our paper. We believe that we have addressed all the concerns. If there is any aspect that you feel has not been fully resolved, we would be happy to provide further information. If you are satisfied with our response, we would truly appreciate your consideration in raising your evaluation score.

---

> > ### Comment · Reviewer_cACW · 2024-11-22
> >
> > I have changed my score correspondingly.

---

> > > ### Author Response · Authors · 2024-12-02
> > >
> > > Thank you for your suggestions for helping us improve our paper. We would like to kindly let you know that we have gathered additional results for GGDA on ImageNet. Please see our global response for the results. We do observe that GGDA continues to maintain significant speedups in this large-scale setting.

---

### Official Review · Reviewer_X7t6 · 2024-10-29

**Soundness:** 3
**Presentation:** 2
**Contribution:** 2
**Rating:** 5
**Confidence:** 3

**Summary:**

The paper “Generalized Group Data Attribution” introduces the GGDA framework, designed to enhance data attribution efficiency by grouping training data points. GGDA aggregates training data into groups instead of handling individual points, significantly improving computational efficiency while maintaining comparable accuracy. It extends popular attribution methods like Influence Functions, TracIn, and TRAK to group-based settings, making them suitable for large-scale datasets.

Extensive experiments on various datasets and models validate GGDA’s performance in tasks like dataset pruning and noisy label detection, demonstrating its effectiveness and scalability. However, the paper could benefit from more experiments on real-world large-scale datasets, as well as a deeper theoretical analysis of the K-Means grouping strategy, which plays a critical role in enhancing attribution efficiency yet lacks detailed discussion in the theoretical framework.

Overall, GGDA shows promise for data attribution, but needs further validation for large-scale dataset use and a deeper theoretical analysis of the K-Means grouping strategy.

**Strengths:**

The paper's experimental section introduces K-Means clustering in gradient space as part of the grouping strategy. This innovative design improves attribution accuracy. The approach demonstrates significant advantages in different attribution tasks, such as dataset pruning and noisy label detection, validating its applicability across various scenarios.

**Weaknesses:**

1. **Absence of Large-Scale Dataset Experiments**: The experiments primarily focus on small to medium-scale datasets, leaving out truly large-scale datasets (e.g., billion-level data). To better demonstrate GGDA’s scalability, future work should incorporate experiments on large-scale datasets and report both computational efficiency and attribution performance in such scenarios.
2. **Lack of K-Means Analysis**: K-Means plays a vital role in the proposed method's effectiveness, but the paper lacks theoretical analysis and runtime details for this component. This omission limits the evaluation of its feasibility and efficiency. Providing more detailed descriptions in the appendix or code repository would enhance reproducibility for researchers.

**Questions:**

1. Do the authors plan to conduct experiments on large-scale datasets (e.g., billion-level data) in future work to further validate the scalability and attribution performance of GGDA?
2. The paper lacks details on K-Means runtime and theoretical analysis. The authors could add these implementation details in the appendix to facilitate reproducibility, especially considering that K-Means in the gradient space can be time-consuming when the gradient space is large.

---

> ### Author Response · Authors · 2024-11-22
> **Rebuttal by Authors**
>
> Thank you for your thoughtful review of our paper. Please see our response to the weaknesses and the questions below.
>
> > W1: large scale experiments
>
> We are actively working on running experiments on ImageNet. This requires significant computational resources and running time. We will update the paper with these results as soon as they are available.
>
> > W2: Lack of K-Means analysis and runtimes.
>
> Yes, we totally agree that an analysis of K-means runtime and providing more details about the code will enhance the paper, especially for reproducibility. Our K-means is based on a public implementation https://github.com/subhadarship/kmeans_pytorch. Our code is available in the supplementary material.
>
> In analyzing computational costs, we would like to first highlight the distinction between two types of costs for any attribution method:
>
> 1. The offline training time cost, which refers to all computational overhead incurred during the preprocessing and model preparation phase. This includes data processing, model training, and any preparatory steps needed before deployment. Many retraining-based attribution methods like TRAK inherently have significant training time costs, as they require training multiple model checkpoints for ensemble.
>
> 2. The online serving time cost, which refers to the computational cost incurred during the actual attribution score computation for each test point.
>
> The difference between these two types of costs is that the offline training time cost is incurred only once, and will be amortized over many test points, while the online serving time cost is incurred once for each attribution computation.
>
> The K-means clustering in our approach belongs entirely to the offline training time cost category since it can be performed offline as a preprocessing step. The clustering is completed before seeing any test points and does not impact the online serving. Therefore, we choose to compare the serving time cost of attribution methods for practical consideration.
>
> Nevertheless, we provide a detailed K-means runtime analysis below. Following the notation in [1], we let n be the number of training data points, p be the number of model parameters, d be the (hidden) feature dimension, and m be the number of test data points we want to compute the attribution for. Additionally, we let T be the number of training iterations (for TracIn), and J be the number of checkpoints to be ensembled (for TRAK). The results are shown in the Table below.
>
> |Time Complexity|IF|TracIn|TRAK|
> |-|-|-|-|
> |Original|O(mnp)|O(mnpT)|O(mnpJ) + O(npTJ)|
> |GGDA|O(mKp) + O(Knd)|O(mKpT) + O(Knd)| O(mKpJ) + O(npTJ) +O(Knd)|
>
> We start by explaining IF. Its run time is O(np) for each test data point [1], and O(mnp) in total for the entire test set. The GGDA version of IF will include the K-means computation, but only once. K-means has time complexity O(Knd), but using GGDA can bring down the per test instance time complexity to O(Kp). Therefore, GGDA-IF has time complexity O(mKp) + O(Knd).
>
> Similarly, TracIn has time complexity O(mnpT). TRAK has a serving time complexity O(mnpJ), but also additional model training time O(npTJ).
>
> For all cases, the GGDA runtime will be significantly better when K << n, which is the setting we adopt. Also, as m gets larger, the efficiency gain of GGDA will be further enhanced.
>
> We also show that GGDA is much more efficient empirically even counting K-means time. In the tables below, we show the GGDA total attribution time (column 3) for IF-LiSSA, TracIn, and TRAK. The runtime of the original DA methods is bolded (row 1 with group size 1, i.e., n = K), whereas GGDA can be more than ten times faster even with K-means for larger K.
>
> > IF-LiSSA
> |GroupSize(n/K)|MeanGroupingTime|MeanAttributionTime|MeanTotalTime|
> |-|-|-|-|
> |1|0.00|612.90|**612.90**|
> |4|169.03|194.84|363.88|
> |16|127.63|77.62|205.26|
> |64|92.25|37.82|130.06|
> |256|58.87|29.34|88.21|
> |1024|28.54|27.27|55.80|
>
> > TracIn
> |GroupSize(n/K)|MeanGroupingTime|MeanAttributionTime|MeanTotalTime|
> |-|-|-|-|
> |1|0.00|430.17|**430.17**|
> |4|169.03|134.48|303.51|
> |16|127.63|57.25|184.89|
> |64|92.25|41.02|133.27|
> |256|58.87|36.82|95.69|
> |1024|28.54|35.60|64.13|
>
> > TRAK
> |GroupSize(n/K)|MeanGroupingTime|MeanAttributionTime|MeanTotalTime|
> |-|-|-|-|
> |1|0.00|4317.43|**4317.43**|
> |4|172.80|1463.22|1636.01|
> |16|132.48|617.34|749.82|
> |64|93.77|405.86|499.63|
> |256|57.68|349.92|407.59|
> |1024|28.54|335.64|364.18|
>
> [1] Hammoudeh, Z., & Lowd, D. (2024). Training data influence analysis and estimation: A survey. Machine Learning, 113(5), 2351-2403.
>
> > Remark
>
> Thank you once again for your insightful suggestions and comments, which have been instrumental in enhancing the quality of our paper. We believe that we have addressed all the concerns. If there is any aspect that you feel has not been fully resolved, we would be happy to provide further information. If you are satisfied with our response, we would truly appreciate your consideration in raising your evaluation score.

---

> > ### Comment · Reviewer_X7t6 · 2024-11-25
> >
> > Thank you for your detailed response. I have updated my score accordingly. However, I am still awaiting results from larger-scale experiments.

---

> > > ### Author Response · Authors · 2024-12-02
> > >
> > > Thank you for your patience, for engaging in our paper's discussion and for raising your evaluation score. Please see our global response on ImageNet results. We do observe that GGDA continues to maintain significant speedups in this large scale setting.
> > >
> > > With regards to grouping/kmeans timing, the computation of penultimate layer activation gradients for the entire training set was ~1 hour / 3600 seconds and ~ 175 seconds, 955 seconds, and 3800 seconds for grad-K-Means grouping on group sizes 1024, 256 and 64 respectively. These times are small in comparison to the overall 88,000 seconds runtime for standard DA (group size 1) and may also be considered as pre-processing steps as previously mentioned.
> > >
> > > Regarding evaluation, we are unable to perform retraining based evaluations in time, since training even the ResNet-18 on ImageNet requires significant runtime (>30 hours per retraining). We kindly refer to our ResNet-18 results for CIFAR-10 (Figure 2 in the pdf), where GGDA remains within +/- 0.5% test accuracy w.r.t. DA (lower test accuracy is better in the plot).

---

> > > > ### Comment · Reviewer_X7t6 · 2024-12-03
> > > >
> > > > Thank you for the additional details on efficiency experiments. However, with only these results, I cannot adjust the evaluation score. I look forward to a future version with more comprehensive experiments, including retraining-based evaluations.

---

### Official Review · Reviewer_hzLg · 2024-11-03

**Soundness:** 1
**Presentation:** 2
**Contribution:** 2
**Rating:** 3
**Confidence:** 4

**Summary:**

This paper addresses data attribution estimation, which assesses the contribution of a training sample to a model’s generalization according to a downstream performance metric. While data attribution is beneficial for tasks like data pruning and correcting mislabeled samples, it is often computationally impractical, as its demands scale linearly with the number of training samples. To tackle this, the authors propose Generalized Group Data Attribution (GGDA), which shifts attribution from individual samples to groups of samples. They demonstrate that K-means clustering on activation gradients is an effective heuristic for forming these groups. The authors reframe traditional attribution metrics, including the Leave-One-Out and gradient-based metrics, and apply GGDA to dataset pruning and noisy-label identification in small-scale experiments on MNIST, CIFAR-10, HELOC, and TRAC.

## Claims
1.	GGDA can be applied to any sample-based data attribution method.
2.	It trades attribution fidelity for computational efficiency.
3.	GGDA significantly speeds up data attribution.
4.	It is effective for noisy-label identification and data pruning.
5.	GGDA enables practical applications for large-scale machine learning.

**Strengths:**

The paper is well-written, clearly defining introduced concepts, and is well-motivated, as improving computational efficiency in data attribution is valuable for large-scale machine learning. The authors investigate a generally applicable approach to enhance the computational efficiency of data attribution methods, as claimed. The use a variety of data (tabular, image, text) modalities to validate their approach in downstream supervised learning tasks.

**Weaknesses:**

Weaknesses

1.	The experimental datasets (e.g., MNIST, CIFAR-10) are relatively small, calling into question GGDA’s scalability claims for large-scale ML. Can the method be tested on a larger dataset like ImageNet? Does it maintain an effective compute-fidelity tradeoff as sample size increases?
2.	In Section 4, line 272, the authors claim computational advantages for group data attribution. However, in line 265, they note that “a single batched gradient computation is roughly equivalent in runtime to individual per-sample gradients.” Do the results in Tables 1, 2, and 3 use the best available per-sample data attribution methods? Are implementation of individual per-sample gradient-based methods batched, for example via vmap functionals?
3.	Tables lack clarity regarding ± symbols. Do these indicate multiple trials with different seeds? Are groups recomputed for each trial? Why are the values ±0.0 in Table 1?
4.	Tables 1 and 2 do not include baselines for no data removal.
5.	The rationale for clustering by activation gradient, rather than activations alone, is unclear. Aren’t gradients inherently dependent on activations? Could further intuition be provided?

**Questions:**

1.	It is a bit surprising that group data attribution improves the data selection fidelity. How does GGDA achieve that, and could this improvement be due to the group selection heuristic rather than the method itself?

---

> ### Author Response · Authors · 2024-11-22
> **Rebuttal by Authors**
>
> Thank you for your thoughtful review of our paper. Please see our response to the weaknesses and the questions below.
>
> > W1: large-scale experiments on ImageNet
>
> We are actively working on running experiments on ImageNet. This requires significant computational resources and running time. We will update the paper with these results as soon as they are available.
>
> > W2: individual vs batch gradient speed, and efficient implementation using vmap.
>
> We agree that our statement about batched gradient computation and per-sample gradient computation wasn't clear enough. What we meant was that computing gradients of a batch of B samples is much more efficient than looping through each of those B samples and computing gradients one at a time, because the computation of a larger B is not too much slower than the case of B = 1.
>
> Regarding your suggestion about using efficient implementations like vmap. We were not using it but we really appreciate the suggestion. We indeed found that using vmap can speed up both per-sample gradient and batched gradients, which we illustrate in the table below. For this experiment, we consider the run time (ms) for computing gradients for 8192 training data points in four different cases, where we compare per-sample vs. batch and vmap vs. non-vmap.
>
> | Per-sample Grad | Per-sample Grad (vmap)| Batched Grad  (B = 64) | Batched Grad (vmap) (B = 64)|
> |-|-|-|-|
> | 6760 | 12.5 (540x)| 560 (12x)   | 2.23  (3031x)|
>
> We would like to emphasize that 1) Although we were not using vmap, our comparison in the paper was fair because all implementations didn't use it 2) Optimizations like vmap are orthogonal to what we are proposing and can be combined with our GGDA method as well.
>
> We will consider rerunning our experiments with vmap in our next version. We really appreciate the suggestion.
>
> > W3: Table ± symbols
>
> Thank you for seeking clarification here. The ± symbols in the tables do indeed indicate multiple training trials across 10 different seeds, while keeping groups and attributions fixed. Table 1 presents Logistic Regression (LR) results, where there is no variation across training trials due to convexity (hence, ± 0.0). We have updated the table captions to clarify these points.
>
> W4: Add baselines to pruning tables.
>
> Thank you for the suggestion. The baseline of no removal is added to both Table 1 and Table 2, where we see a clear improvement with GGDA.
>
> > W5: Rationale for clustering by activation gradient vs activations
>
> We investigate both clustering strategies and find that gradient k-means outperforms activation k-means in most cases. The effectiveness of gradient k-means can be attributed to that it naturally aligns with how attribution values are calculated for most methods, particularly influence functions. This alignment helps ensure that points grouped together are likely to have similar attribution values.
>
> However, this alignment is not guaranteed, which is why we investigate multiple grouping strategies in our work. The key insight is that an ideal grouping strategy should cluster together points with similar attribution values. The degree of this similarity can vary depending on the specific dataset and attribution method being used.
>
> > Q1: Why does grouping attribution outperform single-point attribution on data pruning?
>
> We hypothesize that GGDA's superior performance in pruning stems from two key factors:
>
> First, while per-sample attributors excel at identifying highly important datapoints, they struggle with accurately estimating the least important points, which is crucial for pruning tasks. This means that while the ranking of datapoint importance is reliable for important points, it becomes less accurate for unimportant ones. We believe this may be due to numerical instabilities in Hessian estimation, particularly affecting influence function methods. The use of groups in GGDA helps mitigate this by smoothing out independent estimation errors across different points.
>
> Second, the effectiveness of GGDA relies heavily on appropriate group selection strategies. For semantically coherent data groups, the attribution values of all points within the groups should be uniform. This makes the identification of coherent data groups crucial for accurate attribution estimation. In contrast, bad grouping creates semantically incoherent sets of samples, making it difficult to estimate accurate attributions for the group as a whole. This highlights why proper group selection is crucial for realizing the benefits of our approach.
>
> > Remark
>
> Thank you once again for your insightful suggestions and comments, which have been instrumental in enhancing the quality of our paper. We believe that we have addressed all the concerns. If there is any aspect that you feel has not been fully resolved, we would be happy to provide further information. If you are satisfied with our response, we would truly appreciate your consideration in raising your evaluation score.

---

> > ### Comment · Reviewer_hzLg · 2024-11-25
> > **Thank you**
> >
> > Thank you, I am waiting for larger-scale experiments to justify more strongly the claims of the paper and to consider raising the score.

---

> ### Author Response · Authors · 2024-12-02
>
> Thank you for your patience and for engaging in our paper's discussion. Please see our global response on ImageNet results. We do observe that GGDA continues to maintain significant speedups in this large scale setting.
>
> With regards to grouping/kmeans timing, the computation of penultimate layer activation gradients for the entire training set was ~1 hour / 3600 seconds and ~ 175 seconds, 955 seconds, and 3800 seconds for grad-K-Means grouping on group sizes 1024, 256 and 64 respectively. These times are small in comparison to the overall 88,000 seconds runtime for standard DA (group size 1) and may also be considered as pre-processing steps as previously mentioned to Reviewer X7t6.
>
> Regarding evaluation, we are unable to perform retraining based evaluations in time, since training even the ResNet-18 on ImageNet requires significant runtime (>30 hours per retraining). We kindly refer to our ResNet-18 results for CIFAR-10 (Figure 2 in the pdf), where GGDA remains within +/- 0.5% test accuracy w.r.t. DA (lower test accuracy is better in the plot).

---

### Official Review · Reviewer_VGjW · 2024-11-04

**Soundness:** 4
**Presentation:** 3
**Contribution:** 1
**Rating:** 3
**Confidence:** 3

**Summary:**

This paper proposes Generalized Group Data Attribution - a method for combining individual data attribution (scores indicating the influence of single training points for single test predictions) into group data attributions (scores indicating the influence of groups of training points for model properties). The resulting attributions are faster to estimate and enable a variety of downstream applications.

**Strengths:**

The paper is clearly written and addresses an important problem, namely the resource-intensive nature of many data attribution methods. The proposed solution is clearly explained, and the writing is clear and concise.

**Weaknesses:**

In my opinion, the main weakness of this paper is the novelty and depth of the investigation. As far as I can tell, the paper proposes turning a point-to-point data attribution method into a group-to-group data attribution method by effectively summing the corresponding individual attributions. This does not seem so fundamental a contribution---e.g., the fact that this reduces sample complexity from O(# points) to O(# groups) seems to follow directly by construction, as without loss of generality one can just call each group a "datapoint."

I think that a more in-depth investigation of the mechanism by which individual attributions are combined could strengthen the paper---for example, are there weighting schemes that improve performance? Are robust estimators (e.g., the median) qualitatively different than taking the average? I also think that the application section could be more fleshed out - the dataset pruning results are the most interesting to me: further investigation into the source of success of GGDA (variance reduction? Soft thresholding? Etc.) would have improved the analysis.

**Questions:**

See weaknesses above. Also - is there any intuition for why grad-k-means works so well as a clustering method?

---

> ### Author Response · Authors · 2024-11-22
> **Rebuttal by Authors**
>
> Thank you for your thoughtful review. Please see our response to the weaknesses and the questions below.
>
> > W1: Limited novelty
>
> We want to clarify a potential misunderstanding here. While using groups instead of individual data points may seem conceptually straightforward, there are significant technical challenges and novel contributions in our work:
>
> First, determining effective grouping strategies is non-trivial. We systematically investigate and compare four different grouping approaches, each with distinct theoretical and practical implications. Such comprehensive analysis of grouping strategies is novel to our knowledge.
>
> Second, extending existing attribution methods to perform well with groups requires careful mathematical consideration and non-trivial algorithmic adjustments. For instance, methods like Influence Functions (IF) need specific modifications for accurate and efficient gradient computation. The complexity lies in properly handling group-level effects while maintaining the theoretical guarantees of the original methods, which we detail in Section 4 along with the modifications needed for other methods like TracIn and TRAK.
>
> Furthermore, a key theoretical insight of our work is that the computational benefits do not come simply from summing individual attributions. Rather, the critical innovation is our ability to compute group attributions directly, reducing complexity from $O(points)$ to $O(groups)$ **without performing intermediate per-sample calculations**. This property holds for IF-style attributions due to their linearity, though notably not for Leave-One-Out (LOO) attributions where group effects cannot be decomposed into sums of individual effects.
>
> > W2: Investigation of combining individual attributions
>
> We appreciate the questions about weighting schemes and robust estimators for combining individual attributions. However, we want to clarify our group attribution process. We don't combine attributions of individual points - instead, we directly compute attributions at the group level. This is a key advantage of our approach, avoiding the need to first compute individual attributions (which incurs significant computational cost).
>
> As derived in our appendix, for IF specifically, the mathematically derived way to compute group attributions is through the summation of individual influences within the group. Importantly, the loss function applied to a group of points should not be mean reduced. The theoretical derivation for TRAK requires more non-trivial and more careful consideration. As such, these are not empirical choices, but follow directly from theoretical foundations.
>
> > W3: Why does GGDA work so well for pruning?
>
> We hypothesize that GGDA's superior performance in pruning stems from two key factors:
>
> First, while per-sample attributors are good at identifying highly important data points, they struggle with accurately estimating the least important points, which is crucial for pruning tasks. This means that while the ranking of data point importance is reliable for important points, it becomes less accurate for unimportant ones. We believe this may be due to numerical instabilities in Hessian estimation, particularly affecting IF methods. The use of groups in GGDA helps mitigate this by smoothing out independent estimation errors across different points.
>
> Second, the effectiveness of GGDA relies heavily on appropriate group selection strategies.
> For semantically coherent data groups, the attribution values of all points within the groups should be uniform. This makes the identification of coherent data groups crucial for accurate attribution estimation. In contrast, bad grouping creates semantically incoherent sets of samples, making it difficult to estimate accurate attributions for the group as a whole.
>
> We believe that further investigation into these aspects presents an interesting direction for future research.
>
> > Q1: Why does grad-k-means work so well?
>
> The effectiveness of gradient k-means can be attributed to it naturally aligning with how attribution values are calculated for most methods, particularly IF. This alignment helps to ensure that points grouped together are likely to have similar attribution values.
>
> However, this alignment is not guaranteed, which is why we investigate multiple grouping strategies in our work. The key insight is that an ideal grouping strategy should cluster together points with similar attribution values. The degree of this similarity can vary depending on the specific dataset and attribution method being used.
>
> > Remark
>
> Thank you once again for your insightful suggestions and comments. We are actively incorporating the above explanations to enhance the quality of our paper. If there is any aspect that you feel has not been fully resolved, we would be happy to provide further information. If you are satisfied with our response, we would truly appreciate your consideration in raising your evaluation score.

---

> > ### Comment · Reviewer_VGjW · 2024-11-26
> > **Response**
> >
> > Thanks to the authors for their responses, and await the new experiments. I still am not entirely convinced that the theoretical analysis is revealing much new insight, but I also don't think that every paper needs to have a deep theoretical component - I just want to understand conceptually whether/how this is different from just treating each cluster of examples as if each were just a single "large" example (after all, the loss function decomposes linearly into individual loss terms).
> >
> > To make things more concrete---suppose we were in a language modeling setting, this feels like just going from token-level to document-level attributions, which is clearly going to be more computationally efficient. Is the key just that the authors have proposed to cluster the examples in a specific way in order to best approximate the individual-level attributions?

---

> ### Author Response · Authors · 2024-12-02
>
> Thank you for your patience and for engaging in our paper's discussion. Please see our global response on ImageNet results. We do observe that GGDA continues to maintain significant speedups in this large scale setting.
>
> With regards to grouping/kmeans timing, the computation of penultimate layer activation gradients for the entire training set was ~1 hour / 3600 seconds and ~ 175 seconds, 955 seconds, and 3800 seconds for grad-K-Means grouping on group sizes 1024, 256 and 64 respectively. These times are small in comparison to the overall 88,000 seconds runtime for standard DA (group size 1) and may also be considered as pre-processing steps.

---

> > ### Author Response · Authors · 2024-12-02
> >
> > To clarify our contributions further. We generally agree with your observation that moving from individual-level to group-level attributions—such as transitioning from token-level to document-level in a language modeling context—can naturally lead to computational efficiency gains. However, we would like to emphasize that summing individual-level scores does not always best represent the group scores. Specifically, even though the loss function can be linearly decomposed into individual loss terms, the contribution score of a group may not. For example for LOO, leaving one subset out doesn't equal leaving each individual out and then summation, which may overlook certain interactions or context-dependent effects that are better captured when scores are computed directly at the group level.
> >
> > In our approach, rather than relying on a summation of individual attributions, we directly compute the group scores. This methodology allows us to retain the efficiency benefits of group-level computation while, in some cases, providing a more representative attribution for the group as a whole. For instance, by clustering examples based on their semantic similarity, our framework is designed to approximate group-level behavior more effectively, capturing nuances that may be missed by simply aggregating individual scores.
> >
> > We hope this clarification addresses your conceptual question and highlights the value of our approach. Thank you again for your thoughtful engagement with our work—we look forward to incorporating your insights as we refine our manuscript.

---

### Author Response · Authors · 2024-12-02
**Additional ImageNet Results**

As requested, we have performed additional experiments on **ImageNet** to demonstrate GGDAs efficiency claims, using models of increasing size: *ResNet-18*, *ResNet-34*, and *ResNet-50*. Experiments are performed using NVIDIA's H100 80GB HBM3.

**TracIn Attribution Times (in seconds) for 1024 Training Examples:**

| Group Size | ResNet-18 | ResNet-34 | ResNet-50 |
| - | - | - | - |
| 1 | 6.50 | 10.90 | 14.48 |
| 4 | 1.64 | 3.04 | 3.24 |
| 16 | 0.77 | 1.00 | 1.32 |
| 64 | 0.56 | 0.66 | 1.12 |
| 256 | 0.42 | 0.58 | 1.02 |

Grouping datapoints enables a) batched gradient computation and b) batched dot products between gradients and the test property gradient vector. We demonstrate speedups of up to 15x with group size 256.

**Influence (Fisher) Attribution Times (in seconds) for 1024 Training Examples:**

| Group Size | ResNet-18 | ResNet-34 | ResNet-50 |
| - | - | - | - |
| 1 | 12.0 | 17.79 | 22.48 |
| 4 | 5.06 | 8.86 | 10.41 |
| 16 | 3.75 | 6.56 | 7.85 |
| 64 | 3.46 | 6.26 | 7.50 |
| 256 | 3.46 | 6.23 | 7.53 |

Here we demonstrate 3-4x speedups on attribution time in the group setting. We also compute attribution times (in seconds) for the *entire* ImageNet training set for the ResNet-18 model.

| Group Size | TracIn | Influence (Fisher) |
| - | - | - |
| 1 | 88217 | 88231 |
| 4 | 22708 | 21364 |
| 16 | 17529 | 17496 |

We observe the same significant speedups (about 5x faster for group size 16) to support our Hessian speed-up claims for influence based methods. Unfortunately, we are unable to perform evaluations on the resulting attribution scores since this requires retraining multiple models, each utilizing >30 hours of training time.

---

### Meta-Review · Area_Chair_CY2j · 2024-12-18

**Metareview:**

**Summary:** The paper proposes group data attribution to replace individual data attribution for more computationally efficient yet robust training. The method uses k-means clustering and other heuristics to group the data.

**Strengths:** The paper is well-written. The problem and method are clearly described, and the experimental results are clearly presented.

**Weaknesses:** Based on the reviews, the main shortcomings are the novelty, lack of theoretical analysis, and larger-scale experiments. The reviewers claim some of the claimed contributions are already known, and the construction lifts the individual data attribution to groups. Reviewer VGjW claims that this is equivalent to summing the attribution of individual points for each group. In terms of analysis, the paper lacks any guarantees or in-depth study of the k-means step. Finally, the results are mainly presented on smaller models/datasets.

**Decision:** The current version of the paper lacks sufficient novelty, theoretical and runtime analysis, and larger-scale results to be ready for publication. I recommend rejection for the current version.

**Additional Comments On Reviewer Discussion:**

The authors provide additional experimental results and try to address some of the concerns raised by the reviewers. They claim that summing individual-level scores does not always best represent the group scores. However, I am not particularly convinced this is true based on their response. The authors need to clarify this in the paper and maybe perform an ablation.

---

### Decision · Program_Chairs · 2025-01-22

Reject